# CDK12 promotes tumorigenesis but induces vulnerability to therapies inhibiting folate one-carbon metabolism in breast cancer

M. G. Filippone[1], D. Gaglio[2,3], R. Bonfanti[1], F. A. Tucci [4], E. Ceccacci[1], R. Pennisi [1], M. Bonanomi [3], G. Jodice[1], M. Tillhon[1], F. Montani[1], G. Bertalot[1], S. Freddi[1], M. Vecchi[1,5], A. Taglialatela[6], M. Romanenghi[1], F. Romeo[1], N. Bianco[1], E. Munzone [1], F. Sanguedolce[7], G. Vago[4,8], G. Viale[1,8], P. P. Di Fiore [1,8], S. Minucci[1,8], L. Alberghina [3,9], M. Colleoni[1], P. Veronesi[1,8], D. Tosoni [1,10✉] & S. Pece [1,8,10✉]

Cyclin-dependent kinase 12 (CDK12) overexpression is implicated in breast cancer, but whether it has a primary or only a cooperative tumorigenic role is unclear. Here, we show that transgenic CDK12 overexpression in the mouse mammary gland per se is sufficient to drive the emergence of multiple and multifocal tumors, while, in cooperation with known onco-genes, it promotes earlier tumor onset and metastasis. Integrative transcriptomic, metabo-lomic and functional data reveal that hyperactivation of the serine-glycine-one-carbon network is a metabolic hallmark inherent to CDK12-induced tumorigenesis. Consistently, in retrospective patient cohort studies and in patient-derived xenografts, CDK12-overexpressing breast tumors show positive response to methotrexate-based chemotherapy targeting CDK12-induced metabolic alterations, while being intrinsically refractory to other types of chemotherapy. In a retrospective analysis of hormone receptor-negative and lymph node-positive breast cancer patients randomized in an adjuvant phase III trial to 1-year low-dose metronomic methotrexate-based chemotherapy or no maintenance chemotherapy, a high CDK12 status predicts a dramatic reduction in distant metastasis rate in the chemotherapy-treated vs. not-treated arm. Thus, by coupling tumor progression with metabolic repro-gramming, CDK12 creates an actionable vulnerability for breast cancer therapy and might represent a suitable companion biomarker for targeted antimetabolite therapies in human breast cancers.

[1] European Institute of Oncology IRCCS, Via Ripamonti 435, 20141 Milan, Italy. [2] Institute of Molecular Bioimaging and Physiology (IBFM), National Research Council (CNR) Segrate, Milan, Italy. [3] ISBE.IT/Centre of Systems Biology, Piazza della Scienza 4, 20126 Milan, Italy. [4] School of Pathology, University of Milan, Milan, Italy. [5] IFOM, The FIRC Institute for Molecular Oncology Foundation, Via Adamello 16, 20139 Milan, Italy. [6] Department of Genetics and Development, Columbia University Irving Medical Center, New York, NY, USA. [7] Department of Pathology, University of Foggia, Foggia, Italy. [8] Department of Oncology and Hemato-Oncology, Università degli Studi di Milano, 20142 Milano, Italy. [9] Department of Biotechnology and Biosciences, Università di Milano-Bicocca, Piazza della Scienza 2, 20126 Milan, Italy. [10] These authors contributed equally: D. Tosoni, S. Pece. ✉email: daniela.tosoni@ieo.it; salvatore.pece@ieo.it

Breast cancer is a heterogeneous disease at the molecular and clinical level[1]. The recognition of different molecular sub-types of breast cancer, namely luminal (divided into luminal A and B based on proliferation rate), HER2-amplified and triple-negative/basal-type, with different characteristics and clinical behavior[2,3], has been critical for informing patient management. Luminal breast cancers, which represent the largest proportion of all breast tumors (~70–75%) and have a more favorable prognosis compared to the other molecular subtypes, benefit from the use of targeted endocrine therapies due to the expression of hormone receptors. Targeted anti-HER2 therapies are also available for HER2-amplified breast cancers. In contrast, triple-negative breast cancers (TNBCs), lacking hormone receptor and HER2 amplification and accounting for nearly ~15% of all invasive breast tumors, have the poorest prognosis of all subtypes and are commonly treated with standard cytotoxic chemotherapies, in the absence of targeted treatments[4]. Regardless, all the molecular subtypes display heterogeneous behavior in response to therapies and more refined prognostic and predictive biomarkers, as well new therapeutic targets are increasingly in demand.

The cyclin-dependent kinase 12 (CDK12) belongs to the CDK family of serine/threonine protein kinases and has pleiotropic roles in regulating gene transcription, RNA splicing and translation, cell cycle progression, DNA damage response (DDR) and maintenance of genomic stability[5–8]. In transcription, CDK12 controls RNA polymerase II (RNAPII) processivity through phosphorylation of its C-terminal domain (CTD), regulating the initiation to elongation transition[9]. Disruption of this function particularly affects the expression of subsets of genes containing a large number of exons and multiple intronic polyadenylation sites, through premature polyadenylation, transcript shortening and cleavage[6,7]. In particular, CDK12 dysfunction has been linked to defective expression of core DNA damage response (DDR) genes involved in homologous recombination, such as *BRCA1*, *ATR* and *FANC1/2*, and key DNA replication genes involved in G1/S progression[5–8]. Accordingly, CDK12 has been described as a tumor suppressor protein in different types of human cancer, such as high-grade serous ovarian cancer, prostate cancer and TNBC, where CDK12 inactivating mutations are associated with high genomic instability and aggressive clinical behavior[10–14].

Conversely, CDK12 has also been described as a pro-tumorigenic factor in different human cancers[15–17], including breast cancer, where its tumorigenic role appears to depend mainly on its co-amplification and cooperation with the *HER2* oncogene or on its synergistic interaction with oncogenic pathways, such as WNT and IRS1-ErbB-PI3K signaling[18,19]. Therefore, the question remains as to whether CDK12 overexpression per se can induce oncogenic transformation of the mammary epithelium, and whether it has independent clinical and therapeutic significance[20]. This is partly because most investigations on the pro-tumorigenic role of CDK12 employed already transformed breast cancer cell lines, often bearing *CDK12* and *HER2* co-amplification, thus confounding the role of CDK12 as a primary oncogene.

Here, we provide extensive functional evidence that CDK12 is a primary oncogene, which, when overexpressed in the normal mammary epithelium, causes the formation of spontaneous breast tumors, in addition to actively cooperating in chemical carcinogen- or oncogene-induced breast tumorigenesis. By intersecting transcriptomic and metabolic analyses, we discovered a role of CDK12 in the pathological rewiring of cellular metabolism through transcriptional hyperactivation of the serine-glycine-one-carbon (SGOC) network. This metabolic rewiring is required for CDK12-dependent breast tumorigenesis and opens a therapeutic window for the treatment of otherwise therapy-refractory CDK12-overexpressing breast cancers, as

demonstrated in retrospective cohort studies of real-life breast cancer patients and in prospective preclinical investigations in patient-derived xenograft (PDX) models.

## Results

**CDK12 is a primary oncogene in the mammary gland**. CDK12 is expressed at physiological levels in the luminal and basal layers of the normal mammary gland, both in mouse and human (Supplementary Fig. 1a). Therefore, to examine the oncogenic potential of CDK12 in the normal mammary gland, we generated epithelial-specific knock-in mice harboring the human *CDK12* transgene (CDK12-KI) by crossing Cre-loxP conditional *CDK12*-KI mice (CDK12^lox/stop/lox^) with cytokeratin-5 (CK5)-Cre mice (see Methods for details). The use of the basal/stem cell layer CK5 promoter ensured early ectopic CDK12 expression in the entire mammary epithelium, achieving CDK12 expression levels, in both the basal and luminal layer, comparable to those observed in the CDK12-overexpressing human mammary breast cancer model, the BT474 cell line (Supplementary Fig. 1b). Compared with age-matched WT controls, CDK12 overexpression in the mammary epithelium of virgin females induced, by 12–16 weeks of age, a significantly higher frequency of premalignant alterations, such as highly proliferative severe dysplasia and hyperplastic alveolar nodules (HAN)[21,22], indicative of initial stages of tumorigenesis (Fig. 1a, Supplementary Fig. 1c), and of multifocal and multicentric infiltrating carcinoma by 12–24 months of age. These results argue for a direct role of CDK12 overexpression in promoting early appearance of preneoplastic lesions that show marked propensity to progress to advanced stages over time, as expected of a driver oncogene (Fig. 1b, c).

Consistent with this gross morphological evidence, primary mammary epithelial cells (MECs) isolated from CDK12-KI mice generated, in a reconstituted extracellular matrix (Matrigel) assay, outgrowths with typical signs of aberrant morphogenesis, such as filled lumen and multi-lobular invasive branching, in contrast with the hollowed acinar structures generated by normal MECs (Fig. 1d). CDK12-KI MECs also contrasted with their WT counterpart for their ability to escape *anoikis* and survive in anchorage-independent conditions, generating spheroid outgrowths with sustained self-renewal ability over multiple passages: a phenotype associated with malignant transformation of normal cells (Supplementary Fig. 1d)[23]. These in vitro and in vivo findings point to a direct transforming oncogenic potential of CDK12 in the normal mammary epithelium.

**CDK12 overexpression increases tumor onset and multiplicity and enhances metastatic spreading in chemical- and oncogene-driven mammary tumorigenesis**. To study whether CDK12 overexpression also has a cooperative effect in breast tumor formation and progression, we monitored the breast cancer susceptibility of CDK12-KI vs. WT mice in response to the chemical carcinogen, 7,12-dimethylbenz[a]anthracene (DMBA). While in WT mice the appearance of palpable mammary tumors was a rare event, which first appeared after 22 weeks post-treatment, adult CDK12-KI mice exhibited a dramatically high tumor penetrance and multiplicity rate, accompanied by a significantly decreased tumor latency and survival rate (Fig. 1e–g).

We next evaluated the possible synergistic effects of CDK12 overexpression with other oncogenes, by crossing CDK12-KI vs. WT (CK5-Cre) mice with PyMT or NeuN mice, two well-established mouse breast tumorigenesis models. Of note, the latter model is relevant to the possible tumorigenic synergy between CDK12 and HER2, considering their frequent co-amplification in naturally occurring HER2-positive human breast cancers[11,18]. Both CDK12-KI/PyMT and CDK12-KI/NeuN

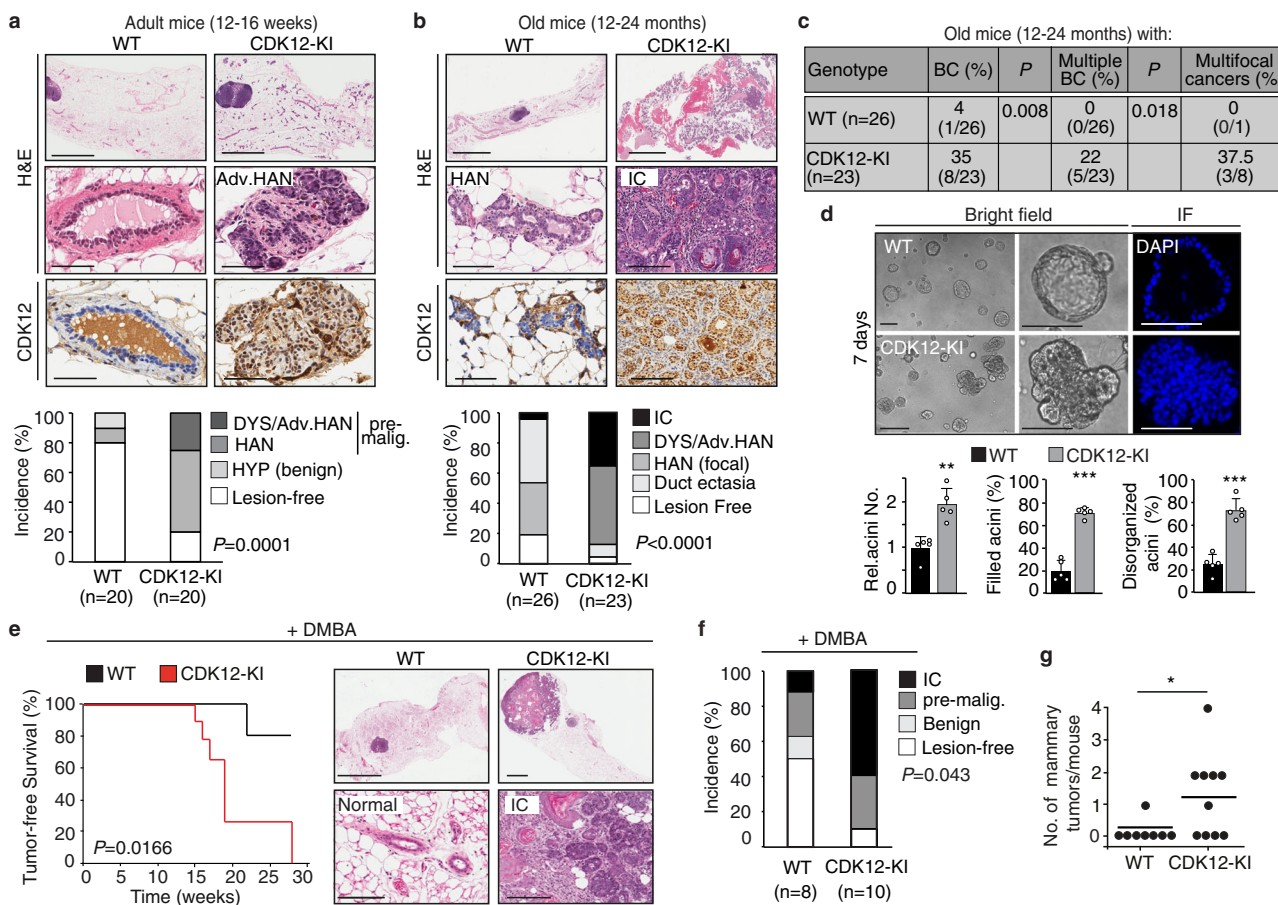

**Fig. 1 CDK12 overexpression enhances spontaneous and carcinogen-induced breast tumorigenesis.** Top, Representative mammary gland histology of age-matched adult (**a**) and old (**b**) female nulliparous CDK12 knock-in (CDK12-KI) vs. wild-type (WT) FVB mice; upper/middle images, hematoxylin and eosin (H&E); lower images, IHC with an anti-human CDK12 specific antibody. Bars (**a**): upper images, 2 mm; middle/lower images, 50 μm; Bars (**b**): upper images, 4 mm; middle/lower images, 100 μm. Bottom, Quantitation of the experiment performed by scoring the highest degree lesion observed in each mouse. HYP, hyperplasia; HAN, hyperplastic alveolar nodule; DYS, dysplasia; Adv. HAN, advanced HAN; pre-malig., pre-malignant; IC, infiltrating carcinoma. *P*, *P*-values, Pearson's test; *n*, number of mice in each group. **c** Frequency (%) of breast cancers (BC) in old CDK12-KI (*n* = 23) vs. WT (*n* = 26) mice as in (**b**). The presence of multiple or multifocal breast cancers is expressed as a percentage of total positive mice. *P*, *P*-values, two-tailed Fisher's test. **d** Top, Typical 3D-Matrigel outgrowth morphology from primary mammary epithelial cells (MECs) isolated from age-matched adult CDK12-KI vs. WT female mice. In immunofluorescence (IF) images, DAPI nuclear staining is shown in blue. Bottom, total number (Rel. Acini No.) and percentage of filled and disorganized acini from CDK12-KI vs. WT MECs. Data are means ± SD. **\**P* = 0.001; **\***P* < 0.001, two-sided unpaired *t*-test (*n* = 5). **e** Left, Kaplan–Meier analysis of tumor-free survival of CDK12-KI (*n* = 11) vs. WT (*n* = 8) mice treated with DMBA (1 mg/dose/week) for 6 weeks. Sacrifice time at the maximum tolerable tumor burden. *P*, *P*-value, log-rank test. Right, Representative histology of DMBA-treated CDK12-KI mice showing an infiltrating carcinoma (IC) vs. WT female mice showing normal/hyperplastic alveoli. Bars, upper panels: right, 1 mm; left 3 mm; lower panels: right, 150 μm; left, 100 μm. **f** Distribution of mammary lesions detected at sacrifice time in CDK12-KI (*n* = 10) vs. WT (*n* = 8) mice treated as in (**e**). Data are as in (**a**, **b**). *P*, *P*-value for incidence of IC + pre-malignant lesions in CDK12-KI vs. WT, two-tailed Fisher's test. **g** Number of tumors in individual CDK12-KI vs. WT mice treated as in (**e**). **\***P* = 0.0286, two-sided unpaired *t*-test. Source data are provided as Source Data file.

female mice showed significantly increased tumor multiplicity and average tumor size, and reduced tumor-free survival rate, compared to their respective WT/PyMT and WT/NeuN controls (Fig. 2a–c, Supplementary Fig. 2a–c). Of note, the reduction in the survival rate was even more evident in CDK12-KI/PyMT vs. WT/PyMT male mice, likely due to the attenuation of the difference in female mice because of their very accelerated kinetics of tumor progression[24]. Regardless, both CDK12-KI/PyMT and CDK12-KI/NeuN female mice displayed a higher propensity to develop multiple spontaneous lung metastases compared to control WT/PyMT and WT/NeuN mice sacrificed at the same endpoint (i.e., primary tumor size ~1.2 cm) (Fig. 2d, e; Supplementary Fig. 2d, e). IHC analysis confirmed that the lung metastases were CDK12-overexpressing (Fig. 2d, Supplementary Fig. 2d). These results demonstrate that CDK12 overexpression significantly enhances early tumor onset, multiplicity and distant

metastatic spreading in NeuN- and PyMT-driven mammary tumorigenesis. These results also argue that CDK12 might establish a broader spectrum of synergistic interactions with known oncogenes in breast tumorigenesis, beyond that already purported for the HER2 oncogene.

**CDK12-induced transformation of normal MECs is associated with specific pathological reprogramming of cellular metabolism.** We modeled CDK12 overexpression in the normal human mammary epithelium by enforcing CDK12 expression in the normal mammary epithelial MCF10A cell line (CDK12-OE MCF10A cells), obtaining CDK12 levels comparable to the breast cancer BT474 cell line that bears CDK12 overexpression[11,18] (Fig. 3a). Compared to control cells, CDK12-OE MCF10A cells displayed: (i) a higher proliferative rate in 2D-culture conditions

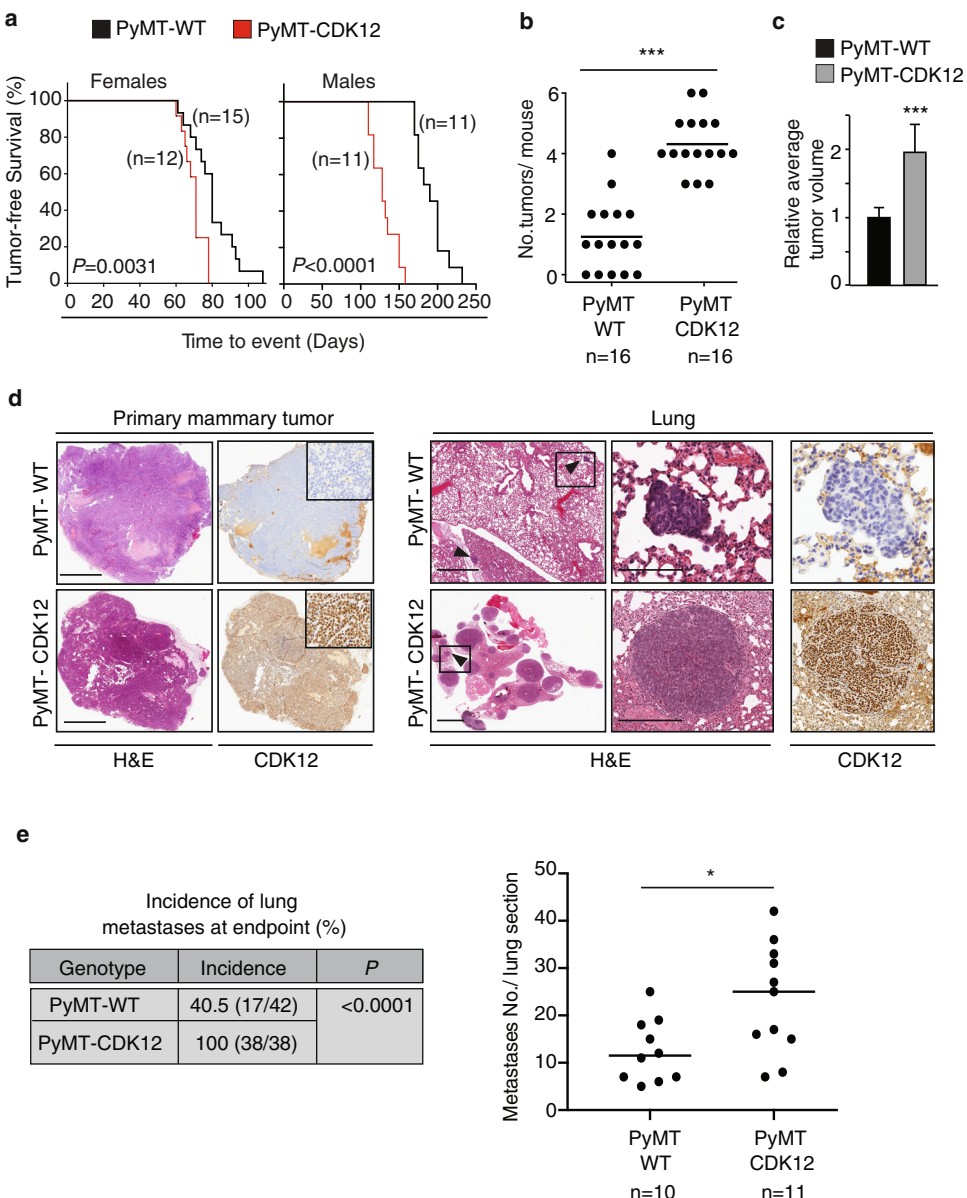

**Fig. 2 CDK12 overexpression increases tumor onset and multiplicity and metastatic spreading in PyMT oncogene-driven mammary tumorigenesis.**
**a** Kaplan–Meier analysis of tumor-free survival of female (left) and male (right) PyMT/CDK12-KI vs. PyMT/WT control mice. All mice are heterozygous for the Cre recombinase transgene. Mice were sacrificed when the maximum tolerable tumor burden was reached (~1.2 cm). The number of mice in the different groups are indicated. Comparison of survival curves, log-rank test. $P = 0.0031$, in Females; $P < 0.0001$, in Males. **b** Distribution of the number of mammary tumors/mouse detected in individual PyMT/CDK12-KI ($n = 16$) vs. PyMT/WT ($n = 16$) mice at a fixed sacrifice time (~5 weeks old). ***$P < 0.001$, two-sided unpaired $t$-test. **c** Relative average tumor volume in PyMT/CDK12-KI ($n = 12$) vs. PyMT/WT ($n = 12$) mice at the time of sacrifice. Data are means ± SD. ***$P < 0.001$, unpaired $t$-test. **d** Representative images of primary mammary tumors (left panels) and synchronous lung metastases (Lung, right panels) in PyMT/CDK12-KI and PyMT/WT mice. Left panels, inserts show a magnification of the tumor areas. Right panels, arrowheads indicate metastatic lesions; boxed areas are magnified on the right. H&E, hematoxylin and eosin staining; CDK12, anti-CDK12 IHC staining. Bars in primary mammary tumor panels, 2 mm; bars in Lung panels: left images, 2 mm; middle/right images: top, 150 µm (PyMT/WT); bottom, 300 µm (PyMT/CDK12-KI). **e** Left, Quantitative analysis of the incidence of lung metastases in PyMT/CDK12-KI vs. PyMT/WT mice. The number of mice in each group is indicated. $P$, $P$-value, two-tailed Fisher's test. Right, quantification of metastases in lung sections from PyMT/CDK12-KI ($n = 11$) vs. PyMT/WT mice ($n = 10$). *$P = 0.018$, two-sided unpaired $t$-test. Source data are provided as Source Data file.

(Fig. 3b); (ii) faulty 3D-Matrigel morphogenesis, with generation of large, filled and multi-lobular invasive outgrowths with sub-verted apical-basal polarity (Fig. 3c, d; Supplementary Fig. 3a); (iii) increased ability to generate spheroids that were larger in size and showed sustained self-renewal ability over multiple passages in anchorage-independent cultures (Supplementary Fig. 3b). Of note, these in vitro CDK12-OE MCF10A cell phenotypes fully mimicked those observed in mouse primary CDK12-KI MECs

(see Fig. 1d and Supplementary Fig. 1c), arguing for the suitability of CDK12-OE MCF10A cells as a tractable model to investigate the molecular events underlying human mammary epithelium transformation by CDK12 overexpression.

We therefore profiled by RNA sequencing (RNA-seq) the transcriptomes of CDK12-OE vs. control MCF10A cells to identify transcriptional programs associated with CDK12 over-expression. KEGG functional analysis of genes upregulated upon

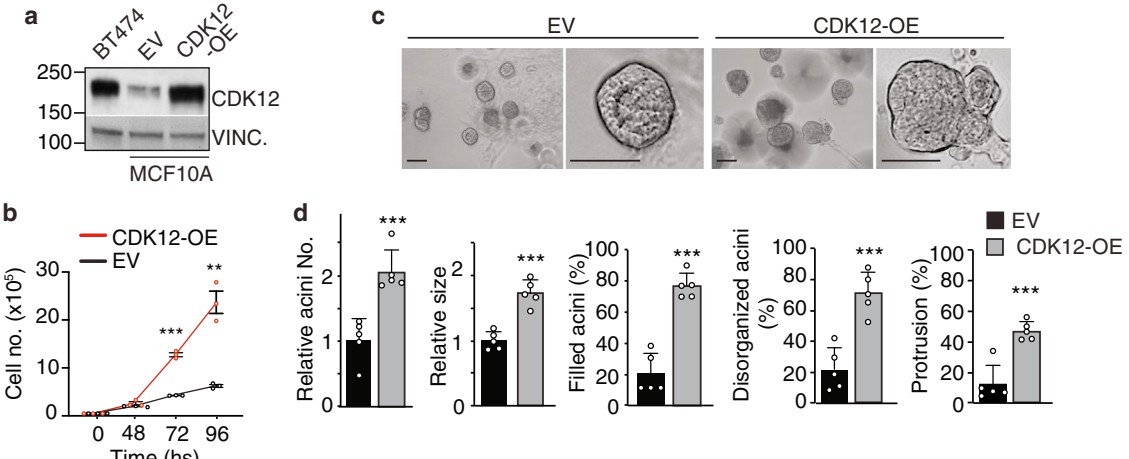

**Fig. 3 CDK12 overexpression induces aberrant functional phenotypes in normal mammary MCF10A cells. a** Immunoblot analysis of CDK12 expression in MCF10A cells stably transfected with CDK12 (CDK12-OE) or empty vector control (EV), and in CDK12-amplified BT474 breast cancer cells. Vinculin (Vinc.), loading control. **b** Growth kinetics of CDK12-OE vs. EV MCF10A cells in 2-D culture. Data are means ± SEM ($n = 3$). **$P = 0.002$, ***$P < 0.001$, two-sided unpaired $t$-test. **c** Typical morphological appearance of 3D-acini generated by CDK12-OE vs. EV MCF10A cells cultivated in Matrigel for 18 days. Left panels in each condition are magnified on the right to better show the hollowed vs. filled/disorganized/branched structures of EV vs. CDK12-OE MCF10A cells, respectively. Bar, 100 μm. **d** Quantitation of the experiment in (**c**). Data are means ± SD ($n = 5$). ***$P < 0.001$, two-sided unpaired $t$-test.

CDK12 overexpression identified various metabolic pathways among the most enriched, including glycine, serine and threonine, citrate (TCA cycle) and pyruvate, folic acid and one-carbon, amino acid, purine, pyrimidine and nucleoside metabolism (Fig. 4a, Supplementary Data 1). The higher abundance, in the transcriptome of CDK12-OE vs. control MCF10A cells, of genes involved in known CDK12-regulated pathways, such as DNA replication, DNA repair, chromatin remodeling, chromosome segregation and cell division[5–7], argued for the suitability of the CDK12-OE MCF10A cell model to study the downstream transcriptional events of CDK12 overexpression (Fig. 4a, Supplementary Data 1). These data clearly suggested that CDK12 overexpression can induce profound modifications of the cellular metabolism. To precisely profile these metabolic alterations, we performed unbiased metabolomics analysis of CDK12-OE vs. control MCF10A cells by liquid chromatography-mass spectrometry (LC-MS). This analysis revealed a metabolic fingerprint unique to CDK12-OE vs. control MCF10A cells, comprised of a set of 46 differentially abundant metabolites (Fig. 4b, Supplementary Data 2 and Supplementary Table 1). Metabolite enrichment and pathway impact analysis of this dataset revealed that CDK12 overexpression influences the glycolysis, glycine and serine synthesis pathway (SSP) and one-carbon (SGOC) metabolism (Fig. 4c), as witnessed by the significantly higher abundance of metabolites associated with these pathways, including serine and glycine, several folate methylated derivatives (5,10-MTHF, 5,10-MeTHF, 5-MTHF), S-adenosyl-methionine (SAM), S-adenosylhomocysteine (SAH), along with an increased glucose consumption and lactate secretion (Supplementary Fig. 3c, d and Supplementary Table 1). In accordance with transcriptomic data, these results further argued for the coordinated and simultaneous hyperactivation of SGOC network pathways in CDK12-OE MCF10A cells.

Transcriptomics and metabolomics data integration analysis of CDK12-OE vs. control MCF10A cells suggested a model in which CDK12 overexpression imposes increased glucose utilization and diversion of the glycolytic flux towards the activation of the SGOC network, coupled with increased downstream activation rate of the methionine cycle (Fig. 4d). Accordingly, key metabolic enzymes involved in serine biosynthesis and one-carbon metabolism were upregulated at the protein and/or mRNA level

in CDK12-OE vs. control MCF10A cells (Supplementary Fig. 3e). Liquid chromatography-mass spectrometry (LC-MS) studies using (U-$^{13}C_6$)-isotope-labeled glucose further confirmed these results showing, in CDK12-OE vs. control MCF10A cells, higher steady-state levels of glucose-derived labeling of intermediates and bioproducts of glycolytic and SGOC metabolism: (i) m + 6 F1,6BP, m + 3 pyruvate and m + 3 lactate (enhanced glycolytic flow to lactate) (Supplementary Fig. 3f); (ii) m + 3 serine and m + 2 glycine (enhanced de novo synthesis through the SSP); (iii) m + 1 5,10 Me-THF, m + 1 5,10-MTHF and m + 1 SAH (enhanced activation of the folate and methionine cycles) (Fig. 4e, Supplementary Fig. 3f); (iv) m + 1 adenosine (purine) and m + 1 thymine (pyrimidine) (enhanced nucleotide biosynthesis) (Supplementary Fig. 3f). Compared to their control counterparts, CDK12-OE MCF10A cells also showed a higher activation of their energy metabolism and de novo synthesis of ATP, as witnessed by the higher steady state levels of: m + 5 ATP, sustained by the enhanced glucose utilization in the pentose phosphate pathway (PPP) to generate ribose 5-phosphate (m + 5 R5P); m + 6 and m + 7 ATP, sustained by the high availability of one-carbon units donated from the SSP and folate cycle (Supplementary Fig. 3f). Metabolic flux analysis confirmed an increased glycolytic flow to lactate and glucose carbon flow into the SSP and SGOC network (Supplementary Fig. 3g, Supplementary Data 3 and Supplementary Table 1). Based on these findings, we reasoned that the profound metabolic adaptations induced by CDK12 overexpression are typical of those enacted by rapidly proliferating cancer cells to meet their increased bioenergetic demand and requirement for nucleotide synthesis.

**CDK12-induced metabolic alterations are required to sustain transformed cellular phenotypes.** We therefore speculated that the metabolic rewiring induced by CDK12 overexpression could represent a selective vulnerability of CDK12-transformed cells to treatments interfering with glycolysis/SGOC metabolism. Consistently, CDK12-OE MCF10A cell proliferation was markedly inhibited by culturing in glucose-free medium or in the presence of the glycolysis inhibitor, 2-deoxyglucose (2-DG), but not in serine-free medium. In contrast, control MCF10A cells were refractory to glucose deprivation or 2-DG treatment, while showing remarkable sensitivity to serine deprivation (Fig. 5a).

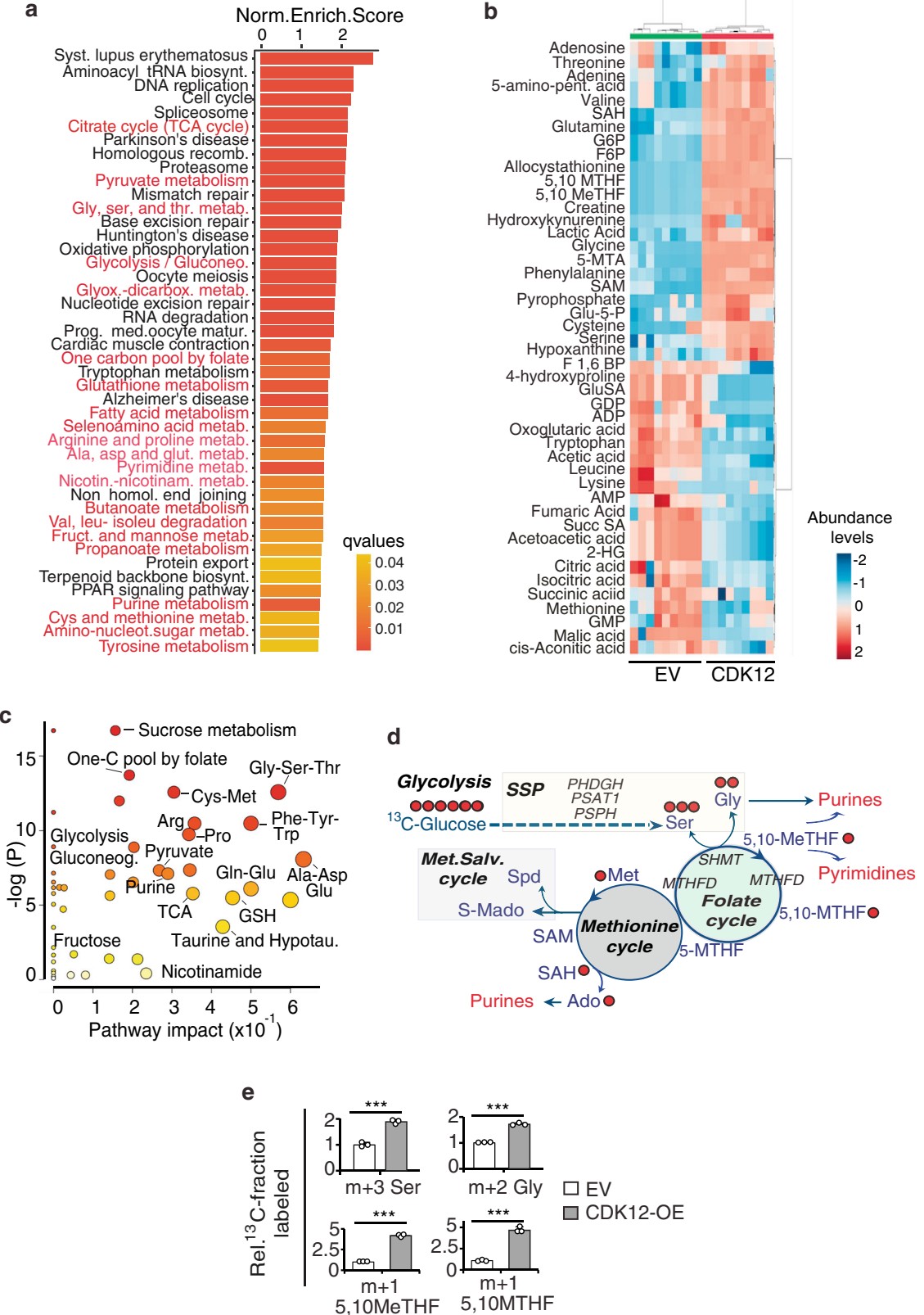

These results point to a highly selective metabolic dependency of CDK12-OE MCF10A cells on glucose as a carbon source to sustain SGOC metabolism, in keeping with metabolomics results.

In line with these data, silencing of the SGOC enzymes, PSAT1 (serine synthesis regulator) and MTHFD1 (folate cycle regulator), significantly inhibited proliferation of CDK12-OE, but not control

MCF10A cells (Fig. 5b, c). Selective inhibition of CDK12-OE vs. control MCF10A cell proliferation was also observed with the CDK12 inhibitor, THZ531[17,25], which fully recapitulated the effects obtained with CDK12 silencing. No synergistic inhibition of CDK12-OE MCF10A cell proliferation was observed combining THZ531 treatment with CDK12 silencing, suggesting that the

**Fig. 4 CDK12 overexpression induces aberrant metabolic phenotypes in normal mammary MCF10A cells. a** KEGG enriched pathways of differentially upregulated genes in RNAseq ($n = 2$) of CDK12-OE vs. EV MCF10A cells. Metabolic pathways are in red. **b** Hierarchical clustering heatmap of the entire set of metabolites differentially expressed in the steady-state profile of CDK12-OE vs. EV MCF10A cells by $t$-test analysis (see abbreviations in Supplementary Table 1). Three independent samples performed in triplicate for each condition were analyzed by LC-MS. The color code scale indicates the normalized metabolite abundance (ranging from −2 up to 2). The clusters containing EV and CDK12-OE MCF10A cells are highlighted in green and red, respectively. **c** Pathway impact analysis of steady-state metabolite profiling as in (**b**). Color intensity (white-to-red) and size of each circle reflects increasing statistical significance, based on the $P$-value [$-\log(P)$] from the pathway enrichment analysis ($y$-axis) and the pathway impact value derived from the pathway topology analysis ($x$-axis), respectively. Gluconeo., gluconeogenesis; TCA, tricarboxylic acid; hypotau., hypotaurine; GSH, Glutatione. **d** Schematic of the metabolic pathways upregulated in CDK12-OE vs. EV MCF10A cells showing derivation and contribution of carbon atoms in intermediate metabolites traced with $^{13}$C-Glucose. Key genes involved in these pathways are also indicated. Red circles, $^{13}$C-labeled carbon atoms. Solid arrows represent direct metabolic reactions, and dashed arrows represent multiple reactions and indirect connections between two metabolites. Outputs of the different metabolic routes are colored in red. SSP, serine synthesis pathway. Met. Salv., methionine salvage. **e** Isotopolog abundance of serine (Ser), glycine (Gly), 5,10 Me-THF, 5,10-MTHF in CDK12-OE vs. EV MCF10A cells grown in medium containing $^{13}$C-Glucose (17.5 mM). Data are the mean ± SD ($n = 3$), represented as percentage isotopomer labeling relative to control condition. ***$P < 0.001$ vs. EV, two-sided unpaired $t$-test. m + n: mass of the isotopomer + n, where n represents the number of heavy carbons ($^{13}$C). Source data are provided as Source Data file.

mechanisms of action of THZ531 depends specifically on its ability to target CDK12 (Fig. 5b). Of note, THZ531 induced transcriptional downregulation of PSAT1 and MTHFD1 in both CDK12-OE and control MCF10A cells, arguing that its selective effects on CDK12-OE MCF10A cells depend on its ability to target their metabolic dependency on the SGOC metabolism (Fig. 5d).

These findings clearly indicate that perturbing folate one-carbon metabolism can represent an effective strategy to reverse the CDK12-induced transformed phenotype. Accordingly, the antimetabolite chemotherapy drug, methotrexate (MTX), which interferes with the dihydrofolate reductase (DHFR) activity[26], efficiently impaired the ability of CDK12-OE MCF10A cells to proliferate in 2D-culture conditions (Fig. 5b) and to generate aberrant outgrowths in 3D-Matrigel (Fig. 5e), while having no effect on control cells (Fig. 5b, e). Notably, MTX had the same inhibitory potency as THZ531 (Fig. 5b, e), which provides further evidence that CDK12-induced transformed phenotypes depend on the pathological regulation of the SGOC metabolism.

To assess the relevance of these findings to CDK12-induced mammary tumorigenesis in vivo, we reverted to the CDK12-KI/PyMT mouse breast tumor model. Mirroring the data obtained in CDK12-OE vs. control MCF10A cells, GSEA showed enrichment of glycolysis and SGOC network genes in the transcriptomic profiles of CDK12-KI/PyMT vs. WT/PyMT tumor cells (Fig. 6a). Likewise, silencing of the two SGOC network genes, PSAT1 and MTHFD1, that appeared transcriptionally upregulated in CDK12-KI/PyMT vs. WT/PyMT tumor cells (Fig. 6a), or silencing of CDK12, selectively inhibited the in vitro proliferation of CDK12-KI/PyMT tumor cells, with no effects on WT/PyMT cells (Fig. 6b, c). Further supporting the therapeutic actionability of the folate one-carbon metabolism for CDK12-overexpressing breast cancers, in vivo MTX administration delayed tumorigenicity and reduced lung metastasis by more than 50% in CDK12-KI/PyMT mice, with no effects observed in WT/PyMT mice (Fig. 7a, b). Opposite effects were observed with paclitaxel (PTX), a standard genotoxic chemotherapy of the taxane family. PTX showed no effect whatsoever on CDK12-KI/PyMT cell proliferation either in vitro or in vivo, while profoundly affecting the growth potential of WT/PyMT tumor cells (Fig. 7c, d). Arguably, the opposite anti-proliferative effects of MTX vs. PTX on CDK12-KI/PyMT and WT/PyMT tumor cells did not depend on intrinsic differences in their basal proliferation rate, which appeared to be similar in both cell types (Fig. 7e). Remarkably, CDK12-KI/PyMT tumors from mice that had progressed on first-line PTX treatment showed a dramatic response to MTX used as a secondary treatment (Fig. 7f). Together, these results point to a scenario in which CDK12 overexpression in breast cancer cells, on the one hand, confers

resistance to standard genotoxic chemotherapy, while, on the other, it introduces a selective therapeutic vulnerability through pathological rewiring of the SGOC metabolism.

**CDK12 overexpression negatively influences prognosis but predicts response to MTX treatment in human breast cancer.** To understand the relevance of CDK12-induced alterations of the SGOC metabolism to naturally occurring human breast cancers, we performed a meta-analysis of transcriptomic data from 1904 patients included in the METABRIC dataset[27]. We analyzed the correlation between CDK12 transcript levels and a SGOC metabolic gene signature (named as SGOC$_{CDK12}$ signature) comprised of 7 CDK12-upregulated genes (PHGDH, PSAT1, SHMT2, MTHFD1, MTHFD1L, DHFR, TYMS). This signature was identified by intersecting the lists of differentially regulated genes in CDK12-KI/PyMT vs. WT/PyMT and CDK12-OE vs. control MCF10A cells with an integrated list of 42 SGOC pathway genes annotated in different public databases (Supplementary Fig. 4a; see Methods for details). In all patients (Spearman's $R = 0.38$, $P < 0.0001$) and across the different molecular subtypes of the METABRIC cohort, we found statistically significant positive correlation of the SGOC$_{CDK12}$ signature with CDK12 transcript levels (Fig. 8a). Next, we analyzed whether the association between CDK12 overexpression and sensitivity to MTX vs. resistance to standard chemotherapy, observed in our model systems in vitro and in vivo, was relevant to real-life breast cancer patients. To do this, we analyzed first the correlation between CDK12 expression and disease course in a retrospective consecutive cohort of ~2400 breast cancer patients[28], with complete clinicopathological follow-up data (median 14.1 years), unstratified or stratified by type of adjuvant systemic therapy. Patients were categorized into three groups based on integrative CDK12 IHC and RT-qPCR screening data analysis (see also Supplementary Information for further details): (i) a first group exhibiting CDK12 protein overexpression over the normal mammary gland by IHC, associated in the vast majority of cases with concomitantly high CDK12 mRNA levels (CDK12$^{HIGH}$ patients, ~23%); (ii) two more groups, both with IHC CDK12 levels indistinguishable from the normal mammary gland (seemingly due to threshold detection limits) which, however, could be clearly distinguished based on high mRNA levels or, vice versa, barely detectable/undetectable mRNA levels, classified as CDK12$^{LOW}$ (~53%) and CDK12$^{NULL}$ (~24%) patients, respectively (Supplementary Fig. 4b, c). We reasoned that our CDK12$^{NULL}$ patients could represent the previously reported fraction of breast cancer patients harboring CDK12 loss-of-function mutations, a condition also described in other cancers,

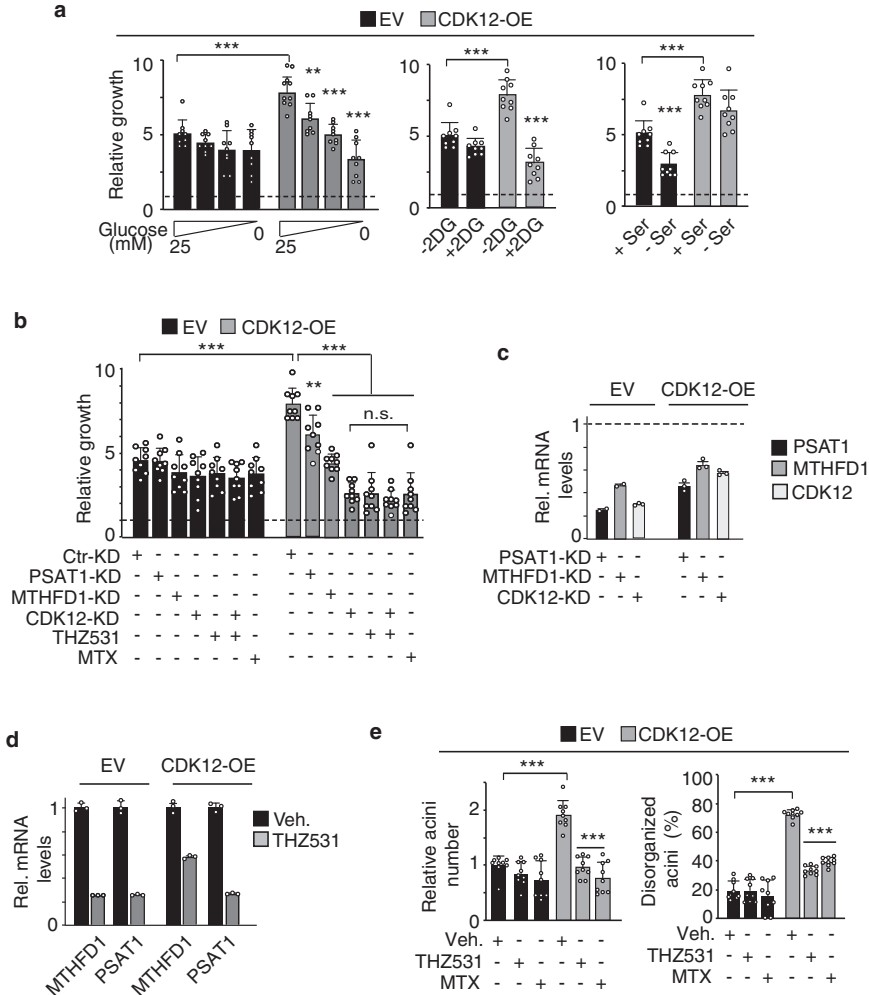

**Fig. 5 CDK12 overexpression induces metabolic dependency on SGOC network pathways in MCF10A cells. a** Three-day proliferation of CDK12-OE and EV MCF10A cells treated with: (i) varying glucose concentrations; (ii) 5 mM 2-deoxyglucose (2DG); (iii) $+/-$ 0.4 mM serine. Data are the mean ± SD ($n = 3$), expressed as relative to day 0 in each condition ($=1$, dashed line). Statistical significance was evaluated for each cell line relative to controls (25 mM, −2DG, + Ser) in each condition, and for control-treated CDK12-OE vs. EV cells in the same experimental condition. $^{**}P = 0.002$; $^{***}P < 0.001$, two-sided unpaired $t$-test. **b** Three-day proliferation of CDK12-OE and EV MCF10A cells silenced for PSAT1 (PSAT1-KD), MTHFD1 (MTHFD1-KD) and CDK12 (CDK12-KD) or treated with THZ531 (100 nM) and MTX (1μM). Data are the mean ± SD, expressed as relative to day 0 in each condition ($=1$, dashed line). Statistical significance, two-sided unpaired $t$-test ($n = 3$): (i) relative to Ctr-KD in each condition; (ii) relative to CDK12-OE cells silenced for CDK12 (CDK12-KD), for CDK12-OE cells treated with THZ531, silenced for CDK12 and concomitantly treated with THZ531, or treated with MTX. $^{**}P = 0.002$; $^{***}P < 0.001$; n.s., not significant. **c** RT-qPCR analysis of knockdown (KD) efficiency in the cells described in (**b**). Data are the mean ± SEM relative to control-silenced cells (Ctr-KD $= 1$, indicated by horizontal dashed line) for each cell line ($n = 1$, performed in technical triplicates). **d** RT-qPCR analysis of the indicated SGOC enzymes in THZ531-treated (100 nM for 48 h) CDK12-OE vs. EV MCF10A cells. Data are the mean ± SEM relative to mRNA levels in control-treated (Veh.) cells (one representative experiment, out of two). **e** Acini formation in 3D-Matrigel by CDK12-OE vs. EV MCF10A cells treated with THZ531 (100 nM), methotrexate (MTX, 1μM), or control-treated (Veh.). Left, data are the mean ± SD, expressed as relative to EV in vehicle-treated condition. Right, data are the mean ± SD and are expressed as percentage of disorganized acini (%) vs. total acini number. Statistical significance, two-sided unpaired $t$-test ($n = 3$) relative to vehicle in each condition, and relative to EV in the same experimental condition for vehicle-treated CDK12-OE cells. Source data are provided as Source Data file.

including ovarian and prostate cancer, and associated with aggressive disease course[10,12,29–31].

This 3-class stratification was clinically relevant: in the unstratified cohort, CDK12[HIGH] and CDK12[NULL] patients showed a significantly higher probability of developing distant metastasis in a multivariable analysis adjusted for clinicopathological parameters (age, tumor size, nodal status, grade, KI67, estrogen/progesterone and HER2 receptor status) compared with CDK12[LOW] patients, both in the entire patient cohort (Supplementary Fig. 4c) and in HER2-negative (HER2-neg) patients (Supplementary Fig. 4d), while no statistically significant differences were observed in HER2-positive

(HER2-pos) patients (Supplementary Fig. 4e), who display in most cases a CDK12[HIGH] status due to HER2 and CDK12 co-amplification[11,18]. These results clearly indicate that a CDK12[HIGH] status represents an independent biomarker of aggressive disease course and poor prognosis in breast cancer, in particular in HER2-neg patients. We also noted that the peculiar poor prognostic behavior of CDK12[NULL] patients is in good agreement with the previously reported association between complete CDK12 loss-of-function and impaired DDR, high genomic instability and aggressive disease course, as observed in high-grade serous ovarian carcinoma and metastatic castration-resistant prostate cancer[8,10,32–34].

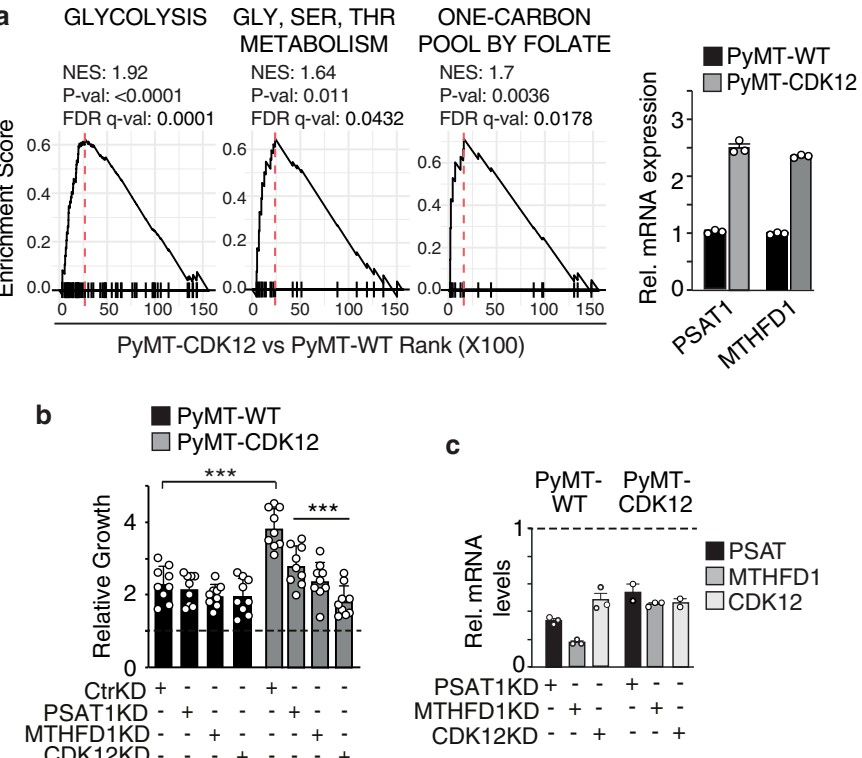

**Fig. 6 CDK12 overexpression induces metabolic dependency on the SGOC network in PyMT/CDK12. a** Left, GSEA of RNA-seq data ($n = 2$) from PyMT-CDK12 vs. PyMT-WT tumor cells, obtained from a pool of 3 different tumors for each strain, showing significant association of CDK12 overexpression with enrichment of gene sets involved in glycolysis (Reactome), glycine-serine-threonine metabolism (Gly-Ser-Thr metabolism, KEGG), one-carbon metabolism (one-carbon pool by folate, KEGG). Right, RT-qPCR analysis of PSAT1 and MTHFD1 in PyMT-CDK12 vs. PyMT-WT cells. mRNA levels are reported relative to control in each condition ± SEM ($n = 1$, out of two independent experiments). **b** Three-day proliferation of PyMT-CDK12 vs. PyMT-WT MECs silenced for PSAT1 (PSAT1-KD), MTHFD1 (MTHFD1-KD) and CDK12 (CDK12-KD). Data are expressed as relative to day 0 in each condition ($=1$, dashed line) and are the mean ± SD ($n = 3$). Statistical significance, two-sided unpaired $t$-test, relative to Ctr-KD in each condition, and relative to WT in the same experimental condition for Ctr-KD PyMT-CDK12 cells. ***$P < 0.001$ relative to matching control. **c** RT-qPCR showing efficiency of PSAT1, MTHFD1 and CDK12 knockdown in PyMT/CDK12 and PyMT/WT MECs. Data are expressed as relative to control-silenced cells (Ctr-KD $= 1$, indicated by horizontal dashed line) for each condition (one experiment performed in technical triplicates).

However, when stratified by type of adjuvant chemotherapy, namely cytotoxic taxane and anthracycline (Tax/AC)-based regimens vs. antimetabolite MTX-based regimens targeting the SGOC network, i.e., CMF (cyclophosphamide, MTX, 5-fluor-ouracil), HER2-neg patients showed different clinical behaviors according to their CDK12 status (the behavior of the different CDK12 groups in response to TaxAC or CMF therapy is described in Supplementary Fig. 5a). We noted that the adverse clinical behavior of CDK12[HIGH] patients treated with Tax/AC chemotherapy, in keeping with our data in model systems, was reverted to a much more favorable disease course in response to adjuvant CMF treatment (Fig. 8b). In particular, in the CDK12[HIGH] group, the 10-year cumulative incidence of distant metastasis dropped from 30.4% in Tax/AC-treated to 12.2% in CMF-treated patients (HR, 4.31, $P = 0.0097$) (Fig. 8b), with a prognosis that became as favorable as that observed in CDK12[LOW] patients. In contrast, no statistically significant differences in response to TaxAC vs. CMF treatment were observed in CDK12[LOW] patients (HR, 1.19, $P = 0.58$), who maintained their favorable prognosis, or CDK12[NULL] patients (HR, 0.76, $P = 0.42$), who maintained a poor prognosis, irrespective of type of chemotherapy (Fig. 8b). These results indicate that CDK12[HIGH] patients show selective sensitivity to CMF therapy compared to CDK12[LOW] and CDK12[NULL] patients, arguing for the suitability of CDK12 as a predictive

therapeutic biomarker for MTX-based therapies in the clinical setting. The reversion of the poor prognostic outcome of CDK12[HIGH] patients, in response to CMF therapy, to a more favorable disease course similar to that observed in CDK12[LOW] patients reveals a typical oncogene addiction-like therapeutic vulnerability in CDK12-overexpressing breast cancers. Definitive evidence of this was provided analyzing a cohort of 213 patients with hormone receptor-negative, node-positive, early breast cancer, of any HER2 status, and with a long-term follow-up (median, 11.3 years), who were randomized in an adjuvant phase III trial (IBCSG 22-00)[35] to low-dose metronomic MTX-based maintenance chemotherapy (cyclophosphamide + MTX, CM) for 1-year or no maintenance chemotherapy at the end of standard adjuvant treatments. Strikingly, a high CDK12 status predicted a ~90% reduction in the cumulative incidence of distant metastasis during the entire follow-up period in the chemotherapy-treated vs. not-treated arm (CDK12[HIGH] $HR_{CM \ vs. \ CTR} = 0.11$, C.I. $= 0.013–0.85$, $P = 0.0065$) (Fig. 8c, Supplementary Fig. 5b). In contrast, in CDK12[LOW] patients, who did not show significant clinicopathological differences compared to CDK12[HIGH] patients (Supplementary Fig. 5b), no additional benefit resulted from the MTX-based therapy as compared to no therapy (CDK12[LOW] $HR_{CM \ vs \ CTR} = 1.1$, C.I. $= 0.051–2.30$, $P = 0.84$). Remarkably, the positive correlation between high CDK12 status and favorable prognosis in the

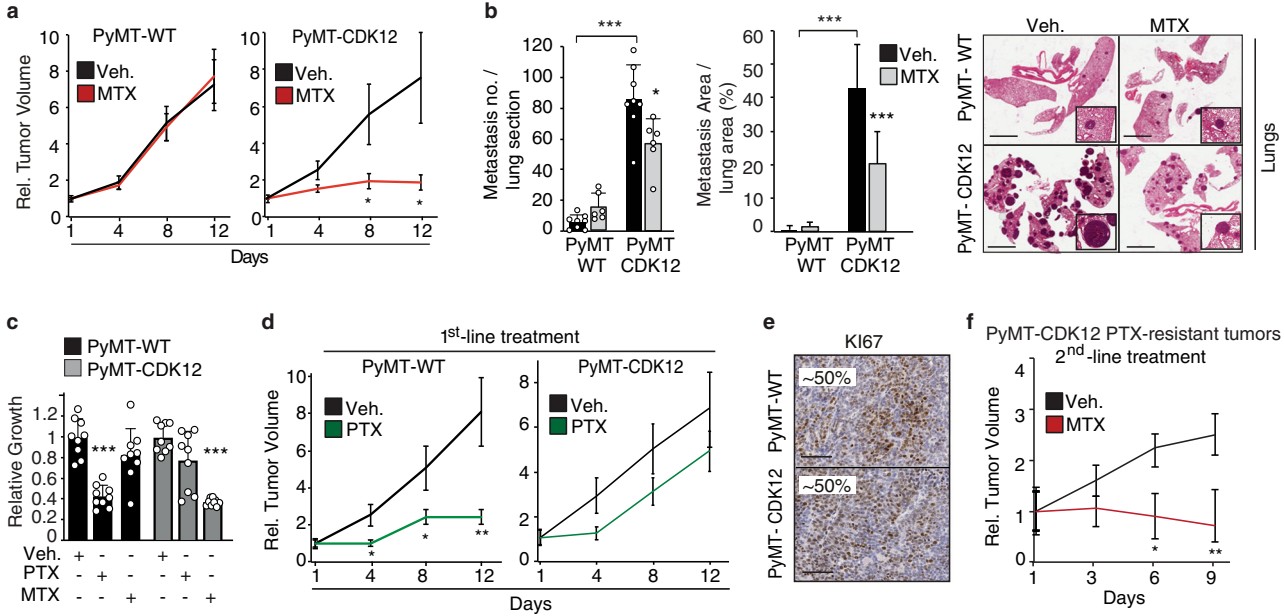

**Fig. 7 CDK12 overexpression induces metabolic vulnerability to methotrexate (MTX) treatment in PyMT/CDK12 mice. a** PyMT/CDK12 vs. PyMT/WT xenograft response to MTX (10 mg/kg) or control vehicle (Veh.) ($n \geq 8$ tumors/condition). Data, expressed as relative to day 1 of treatment, are the mean ± SEM. *$P = 0.019$ (8 days); $P = 0.013$ (12 days) relative to matching control, two-sided unpaired $t$-test (Scatter plots and fitting curves in Source data). **b** Number (left) and total area (middle) of synchronous lung metastases detected at the endpoint in the experiment in (**a**). Data are the mean ± SD; *$P = 0.019$; ***$P < 0.001$, two-sided unpaired $t$-test, relative to vehicle in each condition, and relative to WT in the same experimental condition for vehicle-treated PyMT-CDK12 mice. *$P = 0.019$; ***$P < 0.001$. Left, $n =$ lung sections, WT: Veh. $n = 8$, MTX $n = 6$; CDK12: Veh. $n = 8$, MTX $n = 6$. Middle, $n =$ lung sections, WT: Veh. $n = 14$, MTX $n = 11$; CDK12: Veh. $n = 12$, MTX $n = 13$. Right, Histology showing the lung metastatic burden for each condition. Inserts, magnification of typical metastatic lesions in dashed boxed. Bars, 3 mm. **c** Three-days in vitro growth of PyMT/CDK12 vs. PyMT/WT tumor cells treated with paclitaxel (PTX, 20 nM), or MTX (1 μM) vs. vehicle. Data, expressed as to relative 72 h of treatment of each cell line, are the mean ± SD ($n = 3$). ***$P < 0.001$ vs. matching condition, two-sided unpaired $t$-test. **d** PyMT/CDK12 vs. PyMT/WT xenograft response to first-line PTX (5 mg/kg) and MTX (10 mg/kg) ($n \geq 8$ tumors/condition). Data, expressed as relative to day 1 of treatment, are the mean ± SEM. *$P < 0.05$; **$P < 0.01$ relative to matching vehicle-treated controls, two-sided unpaired $t$-test (Scatter plots and fitting curves in Source data). **e** Representative IHC images for KI67 in PyMT/CDK12-KI vs. PyMT/WT tumor xenografts. The percentage of KI67-positive cells is indicated. Bars, 150 μm. **f** Growth response in PTX-resistant PyMT/CDK12 xenografts from (**d**), randomized to second-line MTX (10 mg/kg) or no treatment (Veh.). Data, expressed as relative to day 1 of treatment ($n \geq 4$ tumors/condition), are the mean ± SEM. *$P = 0.020$; **, $P = 0.003$ relative to vehicle-treated tumors, two-sided unpaired $t$-test (Scatter plots and fitting curves in Source data). Source data are provided as Source Data file.

treated group was independent of other standard prognostic factors, such as tumor size and number of positive lymph nodes in an adjusted multivariable analysis (Supplementary Fig. 5b).

Further evidence of the clinical value of CDK12 was provided by the analysis of three independent cohorts of luminal, TNBC and HER2-pos breast cancer patients treated in the neoadjuvant setting, where the mainstay pre-operative treatment is TaxAC-based therapy. In these patients, we found that a CDK12[HIGH] status invariably predicts refractoriness to TaxAC therapy, independently of standard clinicopathological parameters, testified by a significantly lower rate of complete remission in favor of an increased rate of persistent disease or tumor progression on treatment in CDK12[HIGH] vs. CDK12[LOW] patients (Supplementary Fig. 5d). Strikingly, however, in an independent cohort of TNBC patients who, having resisted neoadjuvant first-line TaxAC therapy, were treated post-surgery with adjuvant CMF in the absence of further therapeutic options, we observed a dramatically reduced 10-year incidence of deaths in CDK12[HIGH] vs. CDK12[LOW] patients (41.9% in CDK12[HIGH] vs. 63.5% in CDK12[LOW], HR 2.55, C.I. = 1.46–4.46, $P = 0.001$) (Fig. 8d, Supplementary Fig. 5e). Therefore, in line with our in vitro and in vivo findings in model systems, the sum of our clinical studies demonstrate that CDK12 overexpression has a dual relevance to human breast cancer, which could help guide clinical decision making: (1) CDK12 represents a biomarker of poor prognosis and aggressive disease course, associated with refractoriness to the

most commonly used chemotherapies (i.e., taxanes, anthracyclines); (2) CDK12 behaves as a predictive biomarker of sensitivity to antimetabolite therapies targeting the SGOC metabolism, such MTX (see also Supplementary Fig. 5e for a schematic overview of the clinical value of CDK12 as a molecular biomarker for patient stratification for prognosis and chemotherapy response prediction).

**Upregulation of SGOC metabolism associates with selective vulnerability in CDK12-overexpressing breast cancer patient-derived xenografts.** To provide preclinical proof-of-concept of the therapeutic actionability of CDK12-induced SGOC metabolic alterations, we generated patient-derived xenografts (PDXs) from CDK12[HIGH] (Patients #1 and #2) and CDK12[LOW] (Patients #3 and #4) human breast cancers (Fig. 9a). The dependency of CDK12[HIGH] breast cancers on CDK12 overexpression was verified by silencing CDK12. This resulted in impaired tumor growth upon orthotopic transplantation, and reduced metastasis and increased survival after intracardiac injection, selectively in CDK12[HIGH], but not CDK12[LOW], PDX-derived cells (Fig. 9a, b; Supplementary Fig. 6a–c). The effects of CDK12 silencing on metastasis were unlikely due to baseline differences in metastatic potential between CDK12[HIGH] and CDK12[LOW] PDXs, since control-silenced Patient #1-CDK12[HIGH] and Patient #3-CDK12[LOW] cells displayed comparable intrinsic propensity for metastatic diffusion (Supplementary Fig. 6c). CDK12 silencing in

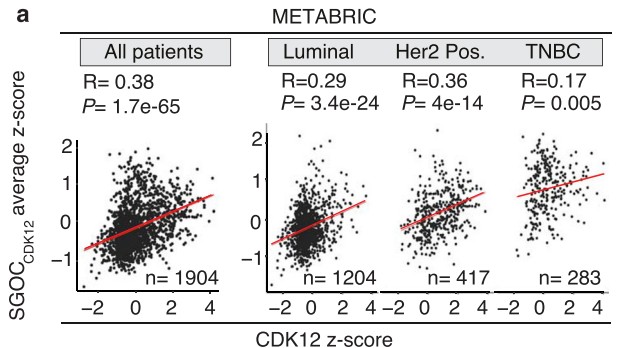

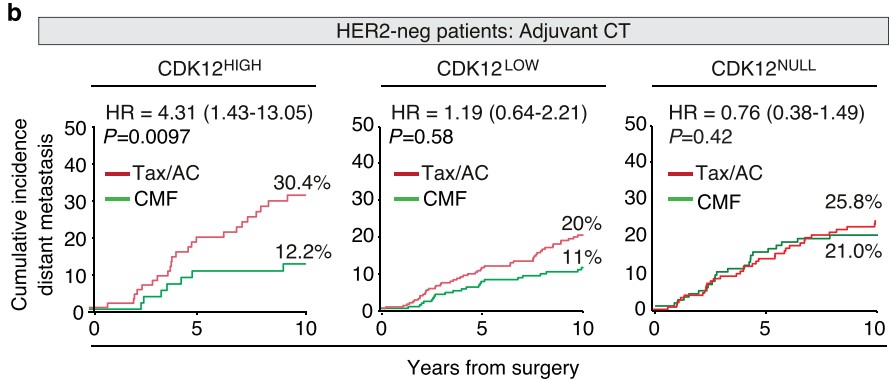

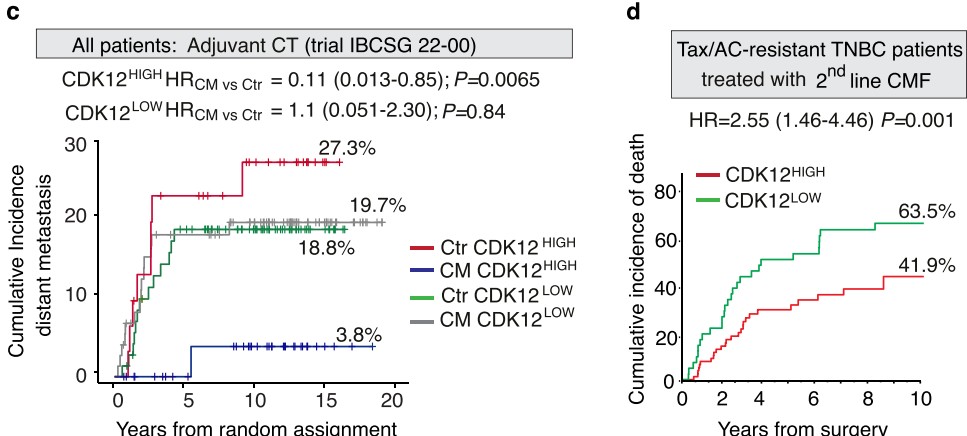

**Fig. 8 CDK12 overexpression is a prognostic and therapeutic biomarker in breast cancer patients and predicts response to antimetabolite therapy targeting the SGOC network. a** Pearson correlation coefficient analysis of the METABRIC breast cancer patient cohort showing positive correlation of a SGOC network gene signature ($SGOC_{CDK12}$ average z-score) with CDK12 transcript levels (CDK12 z-score) in all breast cancers and in the indicated molecular subtypes. Derivation of the $SGOC_{CDK12}$ signature is described in Supplementary Fig. 4. The number of patients in each group is indicated. R, correlation coefficient value; P, P-value. Padj, FDR adjusted P value for the 3 subgroups tested. **b** Cumulative incidence of distant metastasis in CDK12$^{HIGH}$, CDK12$^{LOW}$ and CDK12$^{NULL}$ HER2-negative (HER2-neg) breast cancer patients in response to chemotherapy (CT) with taxane/anthracycline (Tax/AC)- vs. methotrexate-based (CMF, cyclophosphamide, MTX, fluorouracil). The 10-year cumulative distant metastasis rate (%) and the patient number of each class are indicated. HR, multivariable hazard ratio adjusted for standard clinical parameters (see Methods). P, P-value, stratified Wald test. **c** Cumulative distant metastasis in CDK12$^{HIGH}$ and CDK12$^{LOW}$ hormone-negative, lymph node-positive patients randomized to metronomic cyclophosphamide +methotrexate (CM) chemotherapy vs. no chemotherapy (control, Ctr). Patients no.: CDK12$^{HIGH}$ = 31, CDK12$^{LOW}$ = 73 in the control group; CDK12$^{HIGH}$ = 35, CDK12$^{LOW}$ = 74 in the CM therapy group. HR, hazard ratio (see also Supplementary Fig. 5c for cohort description and multivariable analyses). P, P-value, stratified likelihood-ratio test. **d** Cumulative incidence of deaths in CDK12$^{HIGH}$ vs. CDK12$^{LOW}$ neoadjuvant triple-negative breast cancer (TNBC) patients treated with second-line CMF after failure of first-line Tax/AC chemotherapy. HR, multivariable hazard ratio for CDK12$^{HIGH}$ vs. CDK12$^{LOW}$ patients. The 10-year cumulative death rate (%) for each category is indicated. Patient no., CDK12$^{HIGH}$ = 82, CDK12$^{LOW}$ = 43, see also Supplementary Fig. 5e). HR, multivariable hazard ratio; P, P-value, stratified Wald test. Source data are provided as Source Data file.

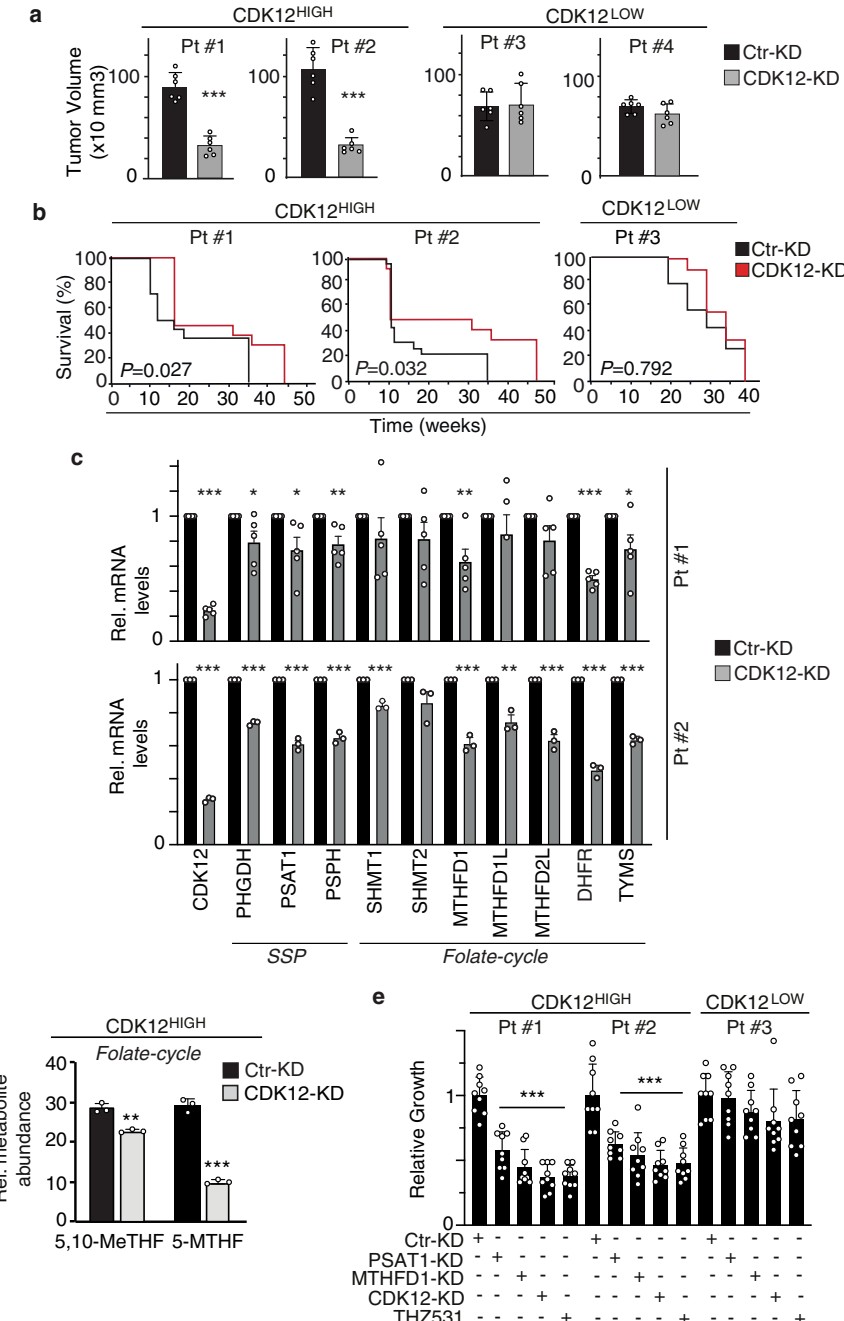

**Fig. 9 Alterations of SGOC metabolism are relevant to real life CDK12^HIGH human breast cancers. a** Two CDK12^HIGH (Pt#1 and #2) and two CDK12^LOW (Pt#3 and #4) breast cancer patient-derived xenografts (PDXs) were silenced with a lentiviral CDK12 shRNA or with a control shRNA (Ctr), and injected orthotopically into the mammary fat pads of NGS mice. Tumor volume was measured at the endpoint (4–6 weeks post-injection). Bars represent the mean ± SD (6 mice per experimental group). ***P < 0.001 vs. Ctr-KD cells, two-sided unpaired t-test. **b** Kaplan–Meier survival analysis of mice injected intracardiacally with CDK12^HIGH Pt#1 and Pt#2, or CDK12^LOW Pt#3 PDX cells treated as in (**a**). Individual mice were sacrificed when signs of distress or severe debilitation became manifest. Number of mice/experimental group: PT#1, 14 for Ctr-KD and 13 for CDK12-KD; Pt#2, 23 for Ctr-KD and 12 for CDK12-KD; Pt#3, 14 for Ctr-KD and 14 for CDK12-KD. P, P-values, by log-rank test. **c** RT-qPCR analysis of the indicated SGOC genes in CDK12^HIGH PDX cells (Pt#1 and Pt#2) silenced or not for CDK12 as in (**a**). Data are expressed as relative to control shRNA (Ctr-KD = 1) for each condition. Bars represent the means ± SEM (n = 5, Pt#1; n = 3, Pt#2). *P < 0.05; **P < 0.01; ***P < 0.001, vs. Ctr-KD, two-sided unpaired t-test. **d** Relative abundance of the indicated folate cycle metabolites in CDK12^HIGH PDX-derived cells control- (Ctr-KD) or CDK12-silenced (CDK12-KD), identified by LC-MS metabolomics. Data are expressed as relative to an internal standard (reserpine) and are means ± SD (n = 3). **P = 0.001; ***P < 0.001, vs. Ctr-KD, two-sided unpaired t-test. **e** Three-day in vitro proliferation of CDK12^HIGH (Pt #1 and #2) vs. CDK12^LOW (Pt #3) cells, control silenced (Ctr-KD) or silenced for CDK12 (CDK12-KD), PSAT1 (PSAT1-KD) and MTHFD1 (MTHFD1-KD), or cultured in the presence of THZ531 (100 nM). Data are expressed as relative to day 3 in each condition and are the mean ± SD (n = 3). ***P < 0.001 relative to Ctr-KD in each condition, two-sided unpaired t-test. Source data are provided as Source Data file.

CDK12[HIGH] PDXs also resulted in the transcriptional down-regulation of enzymes involved at different levels in SGOC metabolism (Fig. 9c). Consistently, upon CDK12 silencing, CDK12[HIGH] PDX cells displayed a significant reduction in the abundance of the folate one-carbon metabolites, 5,10-MeTHF and 5-MTHF, assessed by LC-MS analysis, compared to their control-silenced counterparts (Fig. 9d). We proved the selective metabolic dependency of CDK12[HIGH] PDX cells on the hyper-activation of the SGOC network showing that silencing of the key SGOC pathway enzymes, PSAT1 and MTHFD1, profoundly impaired the in vitro proliferation rate of CDK12[HIGH], but not CDK12[LOW], PDX cells, with an inhibitory effect comparable to that obtained upon CDK12 silencing (Fig. 9e, Supplementary Fig. 6d). Analogous results were obtained with the CDK12 inhibitor, THZ531 which, in line with the effects observed in CDK12-OE MCF10A cells, inhibited PSAT1 and MTHFD1 expression in CDK12[HIGH] breast cancer cells both at the mRNA and protein level (Supplementary Fig. 6e). Together, these data point to the therapeutic potential of targeting SGOC metabolism in CDK12-overexpressing human breast cancers.

**Preclinical validation of MTX as a selective and effective chemotherapy against CDK12-overexpressing PDX breast cancer models.** Our clinical studies in adjuvant and neoadjuvant breast cancer cohorts showed that CDK12 overexpression predicts poor vs. more favorable disease course in patients treated with TaxAC vs. CMF, respectively. Furthermore, evidence from molecular and pharmacological genetics studies in CDK12-KI/PyMT mouse tumors and human breast cancer PDX models revealed that targeting SGOC metabolism is a selective strategy to counteract CDK12-induced tumorigenesis and metastatic spreading. To directly explore the potential of these findings towards clinical applicability, we analyzed the response of CDK12[HIGH] vs. CDK12[LOW] PDXs to MTX and PTX treatment. We selected CDK12[HIGH] and CDK12[LOW] PDXs with a similar basal pro-liferative rate (Supplementary Fig. 7a), to rule out the well-known influence of tumor cell proliferation on vulnerability to chemotherapy[36]. Consistent with previous data, MTX selectively reduced the in vitro cell viability and in vivo tumorigenicity of CDK12[HIGH] PDX cells, while CDK12[LOW] PDX cells were unaffected (Fig. 10a, b). In contrast, CDK12[HIGH] PDX cells were more resistant to PTX, while CDK12[LOW] PDX cells displayed reduced viability upon treatment (Fig. 10a). Of note, silencing of CDK12 selectively restored susceptibility of CDK12[HIGH] PDX cells to PTX, while having no effect on PDX[LOW] cells, confirming the critical role of CDK12 in promoting resistance to taxanes (Fig. 10a).

Taking advantage of a unique set of CDK12[HIGH] and CDK12[LOW] PDXs derived from breast cancer patients who had resisted first-line TaxAC chemotherapy, we showed that CDK12[HIGH], but not CDK12[LOW], TaxAC-chemoresistant PDX cells are still sensitive to the inhibitory effects of MTX in vitro while displaying, as expected, complete refractoriness to PTX (Fig. 10c). Consistent with this data, PSAT1 and MTHFD1 -silencing, and genetic or pharmacological inhibition of CDK12, specifically inhibited proliferation of TaxAC-resistant CDK12[HIGH] tumor cells in vitro, confirming the selective metabolic dependency on SGOC network hyperactivation of CDK12[HIGH], but not CDK12[LOW], TaxAC-resistant cells (Supplementary Fig. 7b). Congruently, mice transplanted with TaxAC-resistant CDK12[HIGH] PDX cells displayed a dramatic tumor growth inhibition in response to MTX treatment (Fig. 10d). These findings strongly argue for the clinical validity of targeting the pathological regulation of the folate-dependent one-carbon metabolism in otherwise therapy-resistant CDK12[HIGH] human

breast cancer patients. Together, the sum of the above in vitro and in vivo studies provides experimental evidence in a preclinical setting of the suitability of MTX as both a first-line and second-line treatment for CDK12[HIGH] breast cancers.

## Discussion

In this study, we provide formal demonstration that CDK12 is a primary oncogenic driver in the normal mammary gland showing that CDK12 overexpression per se: (i) causes the appearance of transformed phenotypes in vitro; (ii) drives the early appearance of preneoplastic lesions and, with a longer latency, the emergence of multiple and multifocal spontaneous infiltrating tumors in vivo; (iii) accelerates in vivo tumorigenesis caused by exposure to the chemical carcinogen, DMBA, and enhances tumor progression and spontaneous distant metastasis in oncogene-driven mouse breast tumor models. Integrative transcriptomic and metabolomic studies revealed a previously uncharacterized interdependency between the activity of CDK12 as a driver oncogene and its ability to orchestrate pathological regulation of cellular metabolism. Indeed, a major consequence of CDK12-induced transformation of normal mammary epithelial cells is the rewiring of glucose metabolism coupled to a hypermetabolic state of distinct pathways integrated in the SGOC network. This is consequent to the simultaneous transcriptional upregulation of several enzymes acting at different levels in the glycolytic/SGOC network, which results in enhanced production of metabolites from the SSP, folate and methionine cycles that characterize the metabolomic profile of CDK12-transformed cells. A remaining question is whether the CDK12 transcriptional regulation of SGOC enzymes is linked to its direct control on RNAPII processivity and mRNA biosynthesis, which would mirror the mechanism described for other CDK12 transcriptional targets, such as core DNA replication and DDR genes[5–9], or rather depends on indirect mechanisms linked to the reported ability of CDK12 to intersect other oncogenic pathways[18,19,37].

Regardless, our functional in vitro and in vivo studies in CDK12-OE breast tumor models and in clinically relevant CDK12[HIGH] human breast cancer PDXs, clearly show that the unique metabolic requirements imposed by CDK12 over-expression are critical for CDK12-induced tumorigenesis, meta-static progression and therapy refractoriness. Indeed, targeting CDK12 activity itself or the glycolytic/SGOC metabolism using molecular genetics or pharmacological approaches led to profound attenuation, or even complete reversion, of CDK12-dependent tumor phenotypes in vitro, and severely curbed tumorigenicity and metastatic spreading in vivo. These results point to the actionability of aberrant CDK12 activity and its downstream hyperactivation of the SGOC metabolism as therapeutic targets for CDK12-overexpressing breast cancers. We provide preclinical proof of this showing that CDK12[HIGH] breast cancer PDXs are selectively susceptible to MTX, an antimetabolite chemotherapy that perturbs folate-dependent one-carbon metabolism. Remarkably, in CDK12[HIGH] breast tumors, MTX was effective both as a first-line treatment and as a second-line treatment for tumors that were otherwise refractory to standard primary genotoxic chemotherapy.

Consistently, in retrospective cohort studies, CDK12[HIGH] status behaved as an overall independent biomarker of poor prognosis in unstratified patients but predicted a considerably more favorable disease course in patient subgroups treated with adjuvant MTX-based chemotherapy, such as CMF, either as a first-line treatment or as a second-line therapy in patients who had no further therapeutic options after failure of primary systemic taxanes and/or anthracyclines. The reversion of the poor prognostic outcome of CDK12[HIGH] patients, in response to

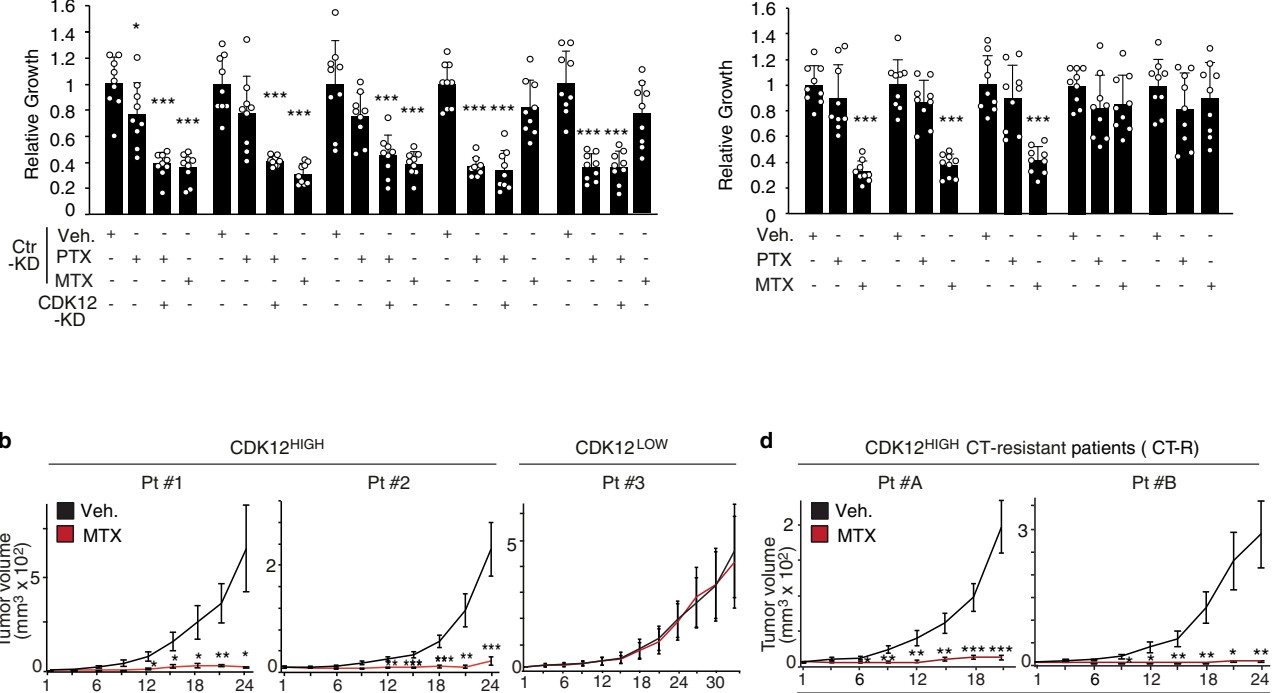

**Fig. 10 Preclinical validation of the therapeutic actionability of SGOC metabolism alterations in CDK12-overexpressing human breast cancers. a** Three-days in vitro growth of CDK12$^{HIGH}$ (Patient #1, #2 and #5) vs. CDK12$^{LOW}$ (Patient #3 and #4) PDX cells control silenced (Ctr-KD) or silenced for CDK12 (CDK12-KD), in response to treatment with paclitaxel (PTX, 20 nM), methotrexate (MTX, 1 μM), PTX plus CDK12-KD, and vehicle (Veh), as a control. Data are expressed as relative to day 3 in each condition and are the mean ± SD ($n = 3$). *$P = 0.035$, ***$P < 0.001$, relative to vehicle in each condition, two-sided unpaired $t$-test. **b** In vivo growth kinetics of CDK12$^{HIGH}$ (Pt#1 and #2) and CDK12$^{LOW}$ (Pt#3) PDXs transplanted in the mammary fat pads of NSG mice administered with vehicle control (Veh.) or methotrexate (MTX, 10 mg/kg) ($n \geq 4$ tumors per experimental group). Data are the mean ± SEM. *$P < 0.05$; **$P < 0.01$; ***$P < 0.001$, relative to vehicle-treated mice, two-sided unpaired $t$-test. **c** Three-days in vitro growth of cells derived from taxanes/anthracycline chemotherapy-resistant (CT-R) CDK12$^{HIGH}$ (Patients #A, #B and #C) and CDK12$^{LOW}$ (Patients #D and #E) PDXs, in response to treatment with paclitaxel (PTX, 20 nM), methotrexate (MTX, 1 μM) or vehicle control (Veh.) for 72 h. Data are expressed as relative to day 3 in each condition and are the mean ± SD ($n = 3$). ***$P < 0.001$ relative to vehicle in each condition, two-sided unpaired $t$-test. **d** In vivo growth kinetics of two tumor xenografts derived from taxane/antracycline-resistant CDK12$^{HIGH}$ PDX cells (CT-R) (Patient #A and #B) transplanted in NSG mice administered with vehicle control (Veh) or methotrexate (MTX, 10 mg/kg) ($n \geq 8$ tumors/per experimental group). Data are the mean ± SEM. *$P < 0.05$; **$P < 0.01$; ***$P < 0.001$, relative to matching controls, two-sided unpaired $t$-test. Source data are provided as Source Data file.

MTX-based antimetabolite therapy, to a more favorable disease course vis a vis the absence of any significant effect of this treatment in CDK12$^{LOW}$ or CDK12$^{NULL}$ patients reveals a typical oncogene addiction-like therapeutic vulnerability in CDK12-overexpressing breast cancers. We therefore propose that a CDK12$^{HIGH}$ status can represent a clinically valuable predictive biomarker to stratify patients who can benefit from anti-metabolite chemotherapy targeting the SGOC network, both in the adjuvant and the neoadjuvant setting. We have provided formal clinical evidence in support of this concept showing that a high CDK12 status associates with a ~90% reduced risk to develop distant recurrence in hormone receptor-negative, lymph node-positive breast cancer patients randomized to metronomic maintenance with a MTX-based chemotherapy or no main-tenance chemotherapy after completion of standard adjuvant treatments, which is a findings of the utmost clinical relevance considering the overall poorer prognosis and the limited ther-apeutic options for this subgroup of patients.

The importance of these findings from a clinical standpoint is only too obvious considering that, while the metabolic vulner-abilities of tumor cells have long represented attractive targets for cancer treatments, the selection of eligible patients based on the actual dependency of their neoplasms on specific metabolic

pathways still remains difficult. Indeed, the heterogeneous clinical efficacy of antimetabolite therapies, such as 5-FU and MTX, despite the ubiquitous presence of their metabolic targets, has contributed to the progressive decline in their use in the clinic, in favor of more modern regimens containing anthracycline and/or taxanes. In this scenario, the reported association between CDK12 oncogenic function and pathological SGOC regulation is a paradigm of how a deeper understanding of the metabolic vulnerabilities specifically underlying the process of malignant transformation by a given oncogene may aid accurate biomarker-based patient stratification for the selection of targeted anti-cancer therapies.

In sum, the findings from this study provide mechanistic insights into the perspective use of CDK12 as a biomarker for more accurate personalized tailoring of both neoadjuvant and adjuvant treatments in breast cancer.

## Methods

**Study design.** The aim of this study was to provide formal demonstration of CDK12 as a primary driver of oncogenic transformation of the normal mammary epithelium and highlight the molecular workings associated with this event. We started with the analysis of the consequences of transgenic CDK12 overexpression in the mouse mammary gland, followed by transcriptomic and metabolomic profiling of normal mammary epithelial cell models upon enforced CDK12

expression. Next, we employed retrospective consecutive clinical cohorts of adjuvant and neoadjuvant breast cancer patients and performed prospective functional studies in preclinical patient-derived xenograft models to provide evidence that CDK12-induced pathological rewiring of cellular metabolism represents an actionable vulnerability for breast cancer therapy, and that CDK12 represents a suitable companion diagnostic for targeted antimetabolite therapies in naturally occurring CDK12-overexpressing breast cancer. No patients or animals were excluded from analysis. For in vivo mouse experiments, sample sizes were selected based on past experience and published experiments, and experiments included a minimum of three mice per group. Mice were randomly assigned to each experimental group. The number of biological and experimental replicates is described in the figure legends.

## Patient cohort studies

*In silico meta-analysis of the correlation between CDK12 and SGOC gene transcriptional levels in the METABRIC breast cancer patient cohort.* Publicly available RNA expression and associated clinicopathological data of 1904 patients from the METABRIC (Molecular Taxonomy of Breast Cancer Consortium) dataset[27] were retrieved from cBioPortal (http://www.cbioportal.org/). Association between variations in CDK12 transcription levels and activation state of the SGOC metabolism in all METABRIC samples was assessed using a CDK12-derived SGOC metabolic signature (named as $SGOC_{CDK12}$ signature) comprised of 7 CDK12-upregulated genes (DHFR, TYMS, MTHFD1L, MTHFD1, PHGDH, SHMT2, PSAT1). The signature was obtained by intersecting the lists of upregulated genes in CDK12-OE vs. control MCF10A cells and PyMT-CDK12-KI vs. PyMT- WT tumor cells (log2 fold change ≥ 0.5, adjusted P value < 0.05) with an integrated list of 42 genes annotated in SGOC-related pathways in different public pathway databases (KEGG-One Carbon Pool By Folate, KEGG-Folate Biosynthesis, REACTOME-Metabolism Of Folate And Pterines, REACTOME-Serine Biosynthesis). The average of the *z*-scores of log-transformed mRNA expression level (Illumina Human v3 microarray) of each of the 7 genes represents the $SGOC_{CDK12}$ score used for the correlation analyses (Pearson's correlation coefficient) with the CDK12 expression *z*-score of each patient. Statistical analysis were performed in R (http://www.r-project.org, version 4.1).

*Retrospective analysis of the clinical value of CDK12 as a prognostic and therapeutic biomarker in a real-world cohort of adjuvant breast cancer patients.* This analysis was performed using a retrospective consecutive cohort of ~2400 breast cancer patients described in earlier studies[28]. Patient stratification by CDK12 status was based on integration of immunohistochemistry (IHC) and RT-qPCR data available for 1713 patients. IHC revealed high heterogeneity of CDK12 protein expression across breast cancers. This inter-tumor heterogeneity contrasted with the uniform pattern of CDK12 expression in the normal mammary tissue, typically characterized by extremely low or even undetectable CDK12 levels (IHC score ≤1.0).

This permitted to distinguish patients with breast cancers bearing overt CDK12-overexpression (IHC intensity score >1.0) from patients whose tumors displayed CDK12 protein levels indistinguishable from those detected in the normal mammary gland (IHC score ≤1.0). Patients were also distinguished by high or low CDK12 mRNA levels according to a quartile-based distribution of transcript levels: the first quartile corresponding to the low mRNA class and the remaining three quartiles to the high one. Analysis of the relative distribution of IHC vs. RT-qPCR data across the entire cohort revealed that CDK12 protein overexpression at the protein level was almost invariably associated with concomitantly high CDK12 mRNA levels; in contrast, breast tumors with absence of CDK12 protein overexpression, and therefore indistinguishable by IHC from the normal tissue, could heterogeneously display either high or low/undetectable CDK12 transcript levels. This finding points to the suitability of IHC to recognize a CDK12 overexpressed vs. not-overexpressed/normal-like status, while being inadequate to discriminate breast cancers with downregulated CDK12 expression compared to the normal mammary gland (both displaying an IHC score ≤1.0).

Therefore, based on integrative IHC and RT-qPCR data cross-analysis, patients were classified as follows: CDK12$^{HIGH}$, bearing CDK12 protein overexpression (IHC staining score > 1.0); CDK12$^{LOW}$, with absence of CDK12 protein overexpression (IHC score ≤ 1.0), and concomitantly high mRNA levels (RT-qPCR quartiles: from 2nd-4th quartile); CDK12$^{NULL}$, with absence of CDK12 overexpression (IHC score ≤ 1.0) associated with low/undetectable mRNA levels (1st RT-qPCR quartile). This 3-class categorization showed that CDK12$^{NULL}$ patients invariably display poor prognosis and therapy resistance irrespective of the type of chemotherapy; in contrast, the groups of CDK12$^{HIGH}$ and CDK12$^{LOW}$ patients, corresponding to overexpressed and not-overexpressed/normal-like CDK12 status by IHC, respectively, showed remarkable differences in their disease course in response to different types of chemotherapy.

*Retrospective analysis of CDK12 as a biomarker of response to standard cytotoxic chemotherapy in real-world cohorts of neoadjuvant breast cancer patients.* The response to standard first-line neoadjuvant therapy with Tax/Ac was evaluated through the retrospective analysis of three independent cohorts of luminal (*n* = 120), TNBC (*n* = 99) and HER2-pos (*n* = 54) neoadjuvant breast cancer patients. The rate of complete remission vs. rate of persistent disease or tumor progression on treatment were evaluated as a clinical endpoints for the comparison

of CDK12$^{HIGH}$ vs. CDK12$^{LOW}$ patients. An additional cohort of 125 neoadjuvant patients (CDK12$^{HIGH}$ = 82; CDK12$^{LOW}$ = 43) who, having resisted pre-operative first-line Tax/AC, were treated post-surgery with adjuvant CMF in the absence of further therapeutic options, was retrospectively analyzed to explore the therapeutic actionability of CDK12 overexpression with second-line CMF in otherwise chemotherapy-resistant patients.

*Validation study of CDK12 as a predictive biomarker of response to methotrexate-based antimetabolite therapy in the retrospective analysis of a randomized clinical trial.* This study was performed by the retrospective analysis of a cohort of 213 patients who had been enrolled in a past randomized clinical trial (Trial 22-00) of the International Breast Cancer Study Group (IBCSG). The conclusions at the end of the original trial have been reported in earlier studies[35]. This cohort includes a homogeneous group of patients with hormone receptor-negative and lymph node-positive early breast cancer, with any HER2 status, enrolled at the European Institute of Oncology in Milan from November 2000 to December 2012, who received CMF-based therapy in the adjuvant setting, and were subsequently randomized anytime between primary surgery and 56 days after the first day of last course of adjuvant chemotherapy to metronomic maintenance chemotherapy with CM (cyclophosphamide 50 mg/day orally continuously and methotrexate 2.5 mg twice/day orally on days 1 and 2 of every week for 1 year) or to no maintenance chemotherapy[35]. Survival analysis were performed with the R (http://www.r-project.org, version 4.0.2) and JMP (SAS institute, version 14.0) software.

*Cell culture and clinical samples.* Primary mammary epithelial cells were isolated from CDK12-KI and WT mice, from PyMT/CDK12 and PyMT/WT tumors, and from NeuN/CDK12 and NeuN/WT tumors as previously described[38]. In brief, mice were euthanized and the mammary glands or tumors were rapidly collected and mechanically dissociated with blades/scissors; tissues were digested in 5–10 ml of digestion mixture EDM (DMEM/F12 medium supplemented with 1 mM glutamine, 200 U/ml collagenase and 100 U/ml hyaluronidase) and incubated at 37 °C for 4 h. The homogenate was centrifuged at 80 *g* for 5 min at RT and the supernatant was discarded. To capture the largely intact acinar structures and remove dissociated contaminants, homogenate was sequentially filtered through cell strainers of decreasing pore sizes (100, 70, 40, and 20 μm meshes). Organoids were suspended in 2 ml trypsin, incubated for 5 min at RT then trypsin was inactivated by addition of 10 ml DMEM + 10% FBS + 1X Pen/Strep. Cells were collected by centrifugation and plated in adhesion or in 3D-overlay Matrigel in primary medium, composed of mammary epithelial basal medium supplemented with 5 μg/ml insulin, 0.5 μg/ml hydrocortisone, 2% B27, 20 ng/ml EGF and bFGF, and 4 μg/ml heparin (complete MEBM) + 2% serum. Alternatively, single cell suspensions were plated in poly-HEMA treated plates to generate spheroids in MEBM[38].

All tissues, fresh, frozen, or archival formalin-fixed paraffin-embedded (FFPE), were collected via standard operating procedures approved by the Institutional Ethical Board (Protocols N10/14 and UID2817) of the European Institute of Oncology (IEO, Milan, Italy) and informed consent was obtained for all tissue specimens linked with clinical data. To generate PDX models, fresh specimens from CDK12$^{LOW}$ and CDK12$^{HIGH}$ human breast cancers were transplanted in NSG mice as previously described[39]. CDK12 status was evaluated by IHC analysis and RT-qPCR. Human cancer cell lines were obtained from the digestion of CDK12$^{LOW}$ and CDK12$^{HIGH}$ PDXs and cultured in the MEBM complete media in suspension. For growth assays, cells were plated at a density of 10,000–50,000 cells per well in 35-well plates, using 3 replicates per time point and/or condition, and allowed to attach overnight. Cell growth was assessed by cell counting. Day 0 was considered the day of the initial seeding. For the inhibitor studies, the concentration used in each case was as follows: 5 mM 2-deoxyglucose, 1 μM MTX, 20 nM paclitaxel (PTX), 0.4 μM doxorubicin, 100 nM THZ513. Inhibitor or vehicle in fresh media was administered at day 0 and replenished every 3 days. For testing serine dependence, cells were cultured in MEM-based ductal media, with or without 0.4 mM L-serine supplementation. For glucose restriction studies, cells were cultured in modified medium DMEM/F12 without glucose. Methylcellulose assays were performed as previously described[38]. BT474 cells were cultured in the DMEM medium, supplemented with 10% FBS and 2 mM L-glutamine, while MCF10A cell lines were cultured in a 1:1 mixture of DMEM and Ham's F12 medium, supplemented with 5% horse serum, 2 mM L-glutamine, 20 ng/ml human EGF, 100 ng/ml cholera toxin, 10 μg/ml insulin, and 0.5 μg/ml hydrocortisone. All cells were cultured at 37 °C in a humidified atmosphere containing 5% CO2. MCF10A cells infected with stably lentiviral constructs, were selected with 2 μg/ml puromycin.

*Genetically engineered mouse models.* Mice were housed in pathogen-free animal facilities at the Cogentech Mouse Facility located at IFOM (The FIRC Institute of Molecular Oncology, Milan, Italy). Mouse study approval: all mice have been maintained in a controlled environment, at 18–23 °C, 40–60% humidity and with 12-h dark/12-h light cycles, in a certified animal facility. All animal studies were conducted with the approval of Italian Minister of Health and were performed in accordance with the Italian law (D.lgs. 26/2014), which enforces Dir. 2010/63/EU (Directive 2010/63/EU of the European Parliament and of the Council of 22 September 2010 on the protection of animals used for scientific purposes) and EU 86/609 directive and under the control of the institutional organism for animal welfare and ethical approach to animals in experimental procedures (Cogentech OPBA).

The following mouse strains were used: Ubi-CDK12 (Ozgene), K5-Cre (a gift from Dr. Stefano Klingler, Department of Molecular Biology, University of Salzburg), MMTV-PyMT [FVB/N-Tg(MMTV-PyMT)634Mul/J, Jackson Laboratory, Stock #002376], NeuN [FVB/N-Tg(MMTVneu)202Mul/J, Jackson Laboratory, Stock #002374], NSG (NOD/SCID-IL-2R gamma chain-null mice, NOD.Cg-Prkdcscid Il2rgtm1Wjl/SzJ, Jackson Laboratory, stock #005557). CDK12-KI mice were generated by crossing CDK12$^{lox/lox}$ mice (originally in C57/Bl6 strain, by Ozgene, then backcrossed into FVB strain) with CK5-Cre mice[40]. The expression of the human CDK12 transgene is driven by an UbiC promoter, flanked by a loxp-STOP-loxp cassette, and is activated by Cre recombinase leading to the expression of CDK12 in mammary epithelial cells. PyMT-CDK12 and NeuN-CDK12 mice were generated by crossing CDK12-KI mice with PyMT or NeuN mice. PyMT-WT and NeuN-WT mice were generated by crossing CK5-Cre mice with PyMT or NeuN mice. Mice heterozygous for the Cre recombinase were used as controls in the experiments. No phenotypic differences were scored comparing WT and heterozygous Cre recombinase mice. Expression of the Cre-Recombinase and CDK12 allele was confirmed by PCR genotyping, using the following forward and reverse primers: CRE-FW: AACATGCTTCATCGTCGG; CRE-REV: TTCG GATCATCAGCTACACC; CDK12-FW: TGTTCGTGCAAGTTGAGTCCATCC; CDK12-REV: ATGCCAATGCTCTGTCTAGGGGTTG; CDK12-REVmut(F6rev): GCTCTCTGGAAAGAAAACCAGTGCC. NOD/SCID-IL-2R gamma chain-null mice were used as recipient for transplantation experiments. Mice were maintained on FVB genetic background and each genotype was generated from intercrosses from the same colony. Mice were euthanized when criteria for disease burden were reached (including abdominal distension that impeded movement, loss of >15% of body weight, labored breathing, and/or abnormal posture). Female and male animals were used for the experiments, as appropriate.

*In vivo xenograft studies.* For fat pad xenografts, cells were transplanted as previously described[39]. Briefly, $2–5 \times 10^5$ cells were suspended in a 1:1 mixture of ice-cold PBS-Matrigel and injected into the fat pad of NSG mice (12–16 weeks of age). Tumor size was assessed at indicated time points by caliper measurements of length and width, and the volume was calculated according to the formula (length × width$^2$/2). For the chemotherapy treatment studies, tumors were allowed to grow to palpable lesions (~20–40 mm$^3$), then mice were randomized into groups and each group treated intraperitoneally with either PTX (5 mg/kg in PBS) weekly or MTX (10 mg/kg in PBS) or vehicle (PBS) three times per week until mice in the vehicle-treated group had to be sacrificed. No mice were excluded from the analysis.

For intracardiac injection, animals were anesthetized using 3% isoflurane gas induction and maintained at 2% isoflurane on a rodent heating pad. Mice were then placed in dorsal recumbency under anesthesia. Cells $(1–2 \times 10^5)$ were resuspended in 100 ml of DPBS and injected using a 26 G needle. Injection of cells into the arterial circulation was confirmed through a pulsing of blood in the needle upon injection. Water was changed every 3 days. For volume measurement, experiments were designed to detect a 50% change in tumor size with 80% power and a type I error of 5% using the T-test. For the evaluation of metastasis, the isolated bones of a hind limb were fixed in 70% ethanol for 48 h, decalcified in 10% EDTA and embedded in paraffin. Sectioned bones (10 μm thick) and brains were then stained with hematoxylin–eosin. To quantitatively evaluate the formation of bone and brain metastatic lesions, multiple sections were prepared from brains and bones dissected from mice injected intracardially with Ctr-KD and CDK12-KD cells, and metastatic lesions in each section were counted. Quantification of the overall lung metastatic burden was based on the assessment of the number of morphologically evident metastatic lesions or of the total area of metastatic lesions vs. total lung area in a single section of lung tissue by ImageJ software.

*DMBA administration by oral gavage and pathology.* DMBA was dissolved in corn oil to give a 5 mg/ml stock concentration. At age 6–8 weeks, mice were gavaged with 0.2 ml (1 mg/dose/week) DMBA once a week for 6 weeks as described[41]. Gross clinical examinations of mice were performed weekly, for a total of >30 weeks after the first DMBA treatment, to monitor body weight and tumor progression. Mice were sacrificed either at the end of the study or earlier if they displayed significant weight loss, signs of distress (ruffled fur, leg paralysis, etc.) or palpable tumors >1.2 cm in diameter. Tissues including tumors were fixed in 10% phosphate-buffered formalin and embedded in paraffin. Sections were prepared and stained with hematoxylin and eosin and subjected to a blind review by a pathologist. Breast multiplicity was calculated based on the number of tumors per mouse and the presence of multifocal lesions was assessed in each individual mouse with cancer. Tumor incidence was calculated based on the number of mice showing mammary tumors out of the total number of mice in the group. In a second study, for the analysis of the expression of Ki67, DMBA treatment was started and the mice were analyzed by IHC, ~4 weeks after the end of treatment.

**Materials.** Antibodies were obtained from the following sources: anti-CK5 (ab53121, from Abcam 1:2000 for IHC); anti-CK18 (sc-51582, mouse monoclonal from Santa Cruz Biotechnology, 1:100 for IHC); anti-p63 (DAK-p63, mouse monoclonal from Dako, 1:100 for IHC); anti-Ki67 (SP6, rabbit monoclonal from Thermo Fisher Scientific, 1:200 for IHC); anti-human CDK12 (mouse monoclonal, 1:5000 for IHC on primary human or mouse tissues, 1:30000 for IHC on PDX

tissues, 1:1000 for IB, 1:2500 for IF, stock concentration 1.45 mg/ml) and dual anti-mouse/human CDK12 (mouse monoclonal, 1:3000 for IHC on primary human or mouse tissues, 1:30000 for IHC on PDX tissues, stock concentration 1.51 mg/ml) were produced in-house; anti-Giantin (Poly19243, from BioLegend, 1:500 for IF); anti-Vinculin (V9131, from Sigma, 1:10000 for IB), anti-PSAT1 (ABC950, from Sigma, 1:500 for IF) and MTHFD1 (HPA000704, from Sigma, 1:200 for IB). Secondary antibodies: Alexa Fluor 647 (A-31573) and Alexa Cy3 (715-165-150), from Jackson ImmunoResearch, 1:500 for IF; anti rabbit (7074) and anti-mouse (7076) from Cell Signaling, 1:2500 in IB.

Collagenase, hyaluronidase, insulin, hydrocortisone, S-adenosylmethionine (A2408), 2-deoxyglucose (D6134), glucose (G7528), adenosine (A4036), guanosine (G6264), thymidine (T1895), uridine (U3003), cytidine (C4654), cholera toxin (C8052), L-serine (S4500), THZ531 (A8736), DMBA (D3254), Poly-HEMA, heparin, bovine serum albumin (BSA), Moviol (81381), DAPI (32670), chloroform and formic acid were from Sigma-Aldrich; B27 from Invitrogen; EGF and b-FGF from Peprotech; MEBM from Clonetics; DMEM:F12 w/o amino acid and glucose (MyBioSource.com); HBSS, fetal bovine serum and horse serum from Euroclone; normal donkey serum (NDS) from Jackson ImmunoResearch; methotrexate (MTX), paclitaxel (PTX), doxorubicin from IEO (Milan, Italy); Matrigel growth factor-reduced basement membrane matrix (354234; Becton Dickinson); acetonitrile, methanol, 2-Propanol and water from Honeywell; [U-$^{13}$C$_6$]-Glucose from Cambridge Isotope Laboratories, Inc. All chemicals and solvents used for extraction buffers and for liquid chromatography were LC-MS Chromasolv purity grade.

*Expression and hairpin constructs.* The lentiviral construct harboring the CDK12 protein was engineered by subcloning the human *CDK12* FL cDNA into the BamH1 and Sal1 of pLVX puro lentiviral vector[42]. To delete CDK12, PSAT1, MTHFD1 in different cells, cultures from murine or human tumor cells were infected with lentiviral-shRNAs constructs. PsicoR-shRNA lentiviral vectors were used to abrogate human *CDK12* expression. The following sequence, sh#1: 5′-GGAGCTGAACTCAGTAGGA-3′, was subcloned into HpaI and XhoI sites of the pSicoR vector (gift from Tyler Jacks, Addgene plasmid # 11579; http://n2t.net/addgene:11579; RRID: Addgene_11579). Other lentiviral shRNAs were from Dharmacon and have the following IDs: Mouse CDK12 (TRCN0000023037); mouse PSAT1 (TRCN0000120420), human PSAT1 (TRCN0000035266), mouse MTHFD1 (TRCN0000042028), human MTHFD1 (TRCN0000036526). A hairpin against luciferase was used as control.

*Quantitative RT-qPCR and SDS-PAGE analysis.* Total RNA was extracted with TRIzol (Invitrogen) and then purified using the RNeasy kit (Qiagen). Reverse transcription was performed from 0.02 to 2 μg of total RNA using the High-Capacity cDNA Reverse Transcription Kit (Thermo Fisher) according to the manufacturer's instructions. Quantitative RT-PCR analysis was sometimes performed using the TaqMan™ Fast Advanced Cells-to-CT™ Kit (Thermo Fisher). For the analysis of the patient cohort, total mRNA was extracted from FFPE samples with AllPrep kit (Qiagen) and the cDNAs pre-amplified (10 cycles) before the qPCR reaction. Each target was assayed in triplicate and average values were calculated from triplicate values, when the standard deviation was <0.5, or from the best duplicate values when the standard deviation was ≥0.5. The ΔCt method was used to calculate the mRNA levels of each target gene normalized against housekeeping genes. The $2^{-\Delta\Delta Ct}$ method was used to compare the mRNA levels of each target gene, normalized to the housekeeping genes, relative to an external standard. Taqman Gene Expression Assay IDs are: CDK12 (Hs00212914_m1; Mm00660706_m1), MTHFD1 (Hs01068263_m1; Mm00507092_m1), MTHFD1L (Hs00914916_m1), PSAT1 (Hs01107692_g1; Mm04932904_m1), B2M (Hs99999907_m1;Mm00437762_m1), ACTB (Hs99999903_m1;Mm00607939_s1), GAPDH (Hs99999905_m1; Mm99999915_g1), TBP (Hs00427621_m1; Mm00446973_m1), PSPH (Hs00190154_m1), PHGDH (Hs00198333_m1), SHMT1 (Hs00244618_m1), SHMT2 (Hs01059263_g1), DHFR (Hs00758822_s1), TYMS (Hs00426586_m1), MTHFD2 (Hs00759197_s1), MTHFD2L (Hs01017321_m1).

For SDS-PAGE analysis, cells were lysed in ice-cold RIPA buffer. Samples were clarified by centrifugation and protein content measured using the BCA protein assay kit (Thermo Scientific). Proteins (10 μg) were resolved on 4–15% SDS-PAGE gels and transferred onto nitrocellulose membranes (Bio-Rad Laboratories). Membranes were blocked in Tris-buffered saline (TBS) containing 5% non-fat dry milk and 0.1% Tween 20 (TBS-T), prior to incubation with primary antibodies overnight at 4 °C. The membranes were then washed with TBS-T followed by exposure to the appropriate horseradish peroxidase-conjugated secondary antibody for 45 min and visualized on Kodak X-ray film or with ChemiDoc Imager and acquired by Image Lab Software (Version 5.2.1) using the enhanced chemiluminescence (ECL) detection system (Bio-Rad Laboratories).

*RNA sequencing experiments.* For RNAseq analysis, libraries were prepared using the Illumina TruSeq Stranded Total RNA Gold kit according to the manufacturer's protocol. Sequenced raw reads were processed with NF-CORE RNASeq pipeline (https://github.com/nf-core/rnaseq, version 3.1)[43], using the human GRCh38 and mouse GRCm39 as reference genomes. Transcript abundances were quantified using the Salmon pseudo-aligner[44]. Differential expression of genes was tested with

two-tailed Quasi-Likelihood F Test as implemented in the edgeR R package (version 3.34)[45] with batch correction to account for the different library preparation rounds within replicates. *P* values where FDR adjusted.

Gene Set Enrichment Analysis (GSEA) of the expression data was used to assess enrichment of the KEGG pathways through fgsea package by ranking differentially expressed genes by their *P*-value with the sign of the estimated change in expression level[46].

*Immunofluorescence and immunohistochemistry.* 3D-Matrigel organotypic cultures were generated as previously described[47,48], with minor modifications. In brief, single cell suspensions from bulk primary MECs (~5000 cells/well) or MCF10A cells (~1000 cells/well) were seeded in pure Matrigel-coated 4-well chamber slides (Nunc™ Lab-Tek™ II Chamber Slide™ System, 154526) in a total volume of 700 µl mammary epithelial basal medium containing 2% Matrigel growth factor-reduced basement membrane matrix (354234; BD), supplemented with 2% heat-inactivated FBS or 5% horse serum (MCF10A), 10 ng/ml EGF, 20 ng/ml b-FGF, 4 µg/ml heparin and 2% B27. Cells were incubated at 37 °C in the presence of 5% $CO_2$ for 14 d, with 300 µl fresh medium added every 7 d. The number of organotypic structures was counted 7–14 d after seeding. Resulting organotypic structures were observed and photographed using an inverted phase-contrast microscope equipped with a digital camera. To better visualize hollow- and filled-type structures, the organotypic outgrowths were subjected to whole mount staining with DAPI and anti-Giantin performed directly onto 4-well Lab-Tek chamber slides. Briefly, the 3D-Matrigel organotypic structures were fixed in 4% PFA for 1 h, permeabilized with 0,5% Triton X-100 in PBS for 1 h, and then blocked with a solution containing 5% BSA 10% normal donkey serum 0.2% Triton X-100, 0.05% Tween 20 in PBS for 1 h. The primary antibody was diluted in blocking solution and incubated for 1 day at 4 °C and 1 day at room temperature (RT). The secondary antibody and DAPI were diluted in blocking solution and incubated for 1 day at 4 °C and 1 day at RT. Confocal microscopy analysis was performed using a long-distance objective lens [Leica SP8 Confocal microscope with AOBS, resonant scanner and motorized stage (x, y, z) 25x/0.95 water objective lens]. Immunofluorescence data was collected with LasX (Version 3.5 Leica).

For IHC analysis, 5-µm thick sections of FFPE mammary whole mounts or tumors were assayed with the appropriate antibodies. Slides were digitally scanned with the Aperio ScanScope XT and automatically analyzed with the Aperio ImageScope IHC nuclear algorithm (v12.2.2, Leica Biosystems). Areas containing $3–6 \times 10^4$ cells were analyzed and the percentage of positive cells was calculated. Digital images were processed with Photoshop CS3.

*Metabolomics experiments.* For experiments with cell lines, $2 \times 10^4$ cells/cm$^2$ were plated in 6-well plates with normal growth medium. After 18 h, cells were washed with PBS and incubated for 48 h in fresh DMEM for untargeted experiments or in DMEM supplemented with 17.5 mM [U-$^{13}C_6$]-glucose for targeted experiments. For experiments with PDXs, cells were grown in complete media until the day of the extraction. For the metabolite extraction, cells were quickly washed with NaCl 0.9% and quenched with 500 µl ice-cold 70:30 acetonitrile-water. For cell culture, plates were placed at −80 °C for 10 min, then cells were collected by scraping. For PDXs, samples were placed at −80 °C for 30 min. Samples were sonicated 5 s for 5 pulses at 70% power twice and then centrifuged at 12,000 *g* for 10 min at 4 °C. The supernatant was collected in a glass insert and dried in a centrifugal vacuum concentrator (Concentrator plus/Vacufuge® plus, Eppendorf) at 30 °C for about 2.5 h. Samples were then resuspended with 150 µl of $H_2O$ prior to analyses.

For LC-MS metabolic profiling, LC separation was performed using an Agilent 1290 Infinity UHPLC system and an InfinityLab Poroshell 120 PFP column (2.1 × 100 mm, 2.7 µm; Agilent Technologies). Mobile phase A was water with 0.1% formic acid. Mobile phase B was acetonitrile with 0.1% formic acid. The injection volume was 15 µL and LC gradient conditions were: 0 min: 100% A; 2 min: 100% A; 4 min: 99% A; 10 min: 98% A; 11 min: 70% A; 15 min: 70% A; 16 min: 100% A with 5 min of post-run. Flow rate was 0.2 mL/min and column temperature, 35 °C. MS detection was performed using an Agilent 6550 iFunnel Q-TOF mass spectrometer with Dual JetStream source operating in negative ionization mode. MS parameters were: gas temp: 285 °C; gas flow: 14 l/min; nebulizer pressure: 45 psig; sheath gas temp: 330 °C; sheath gas flow: 12 l/min; VCap: 3700 V; Fragmentor: 175 V; Skimmer: 65 V; Octopole RF: 750 V. Active reference mass correction was through a second nebulizer using masses with m/z: 112.9855 and 1033.9881. Data were acquired from m/z 60–1050. Data analysis and isotopic natural abundance correction was performed with MassHunter ProFinder and MassHunter VistaFlux software (Agilent). Metabolite abundance was expressed relative to the internal standard Reserpine (Supelco) (Relative Metabolite Abundance) and normalized to cell number or protein content depending on the experiments.

The 13-C Metabolic flux analysis ($^{13}$C-MFA) was carried out using INCA v1.7 based on Elementary Metabolite Unit (EMU) framework[49,50]. Flux through metabolic network consisting of Glycolysis, PPP, TCA, FA, & Biomass synthesis was constructed[51] and was estimated by least squares regression of metabolite labelling pattern and measured extracellular fluxes. The flux values of the network were iteratively adjusted using a Levenberg-Marquardt (local search) algorithm to minimize the sum of squared residual (SSR) objective function. The best global fit was found after estimating at least 50 times using random initial guesses for all reactions in the metabolic network. All the fluxes were subjected to chi-square statistical test to assess goodness of fit and 95% confidence intervals were computed[52].

Data used to perform MFA were absolute quantification of glucose and lactate, in spent media, determined enzymatically using YSI2950 bioanalyzer (YSI Incorporated, Yellow Springs, OH, USA), and labeled mass isotopomer distributions using 17.5 mM [U-$^{13}C_6$]-glucose stable isotope tracer. Lactate release and glucose consumption were measured as follows: CDK12-OE and EV MCF10A cells were seeded in a 6-well plate at a density of 0.6 and 0.3 million cells per well in 2 mL culture medium, respectively. After cells had attached overnight, the medium was removed and the wells were rinsed with PBS, followed by the addition of 2 mL fresh medium. Samples of the cell culture supernatant (500 µL) were collected after 48 h and analyzed for glucose consumption and lactate secretion using the YSI2950 bioanalyzer (YSI Incorporated, Yellow Springs, OH, USA). Glucose consumption and lactate release were calculated as follows: consumption = mmol/L of ingredient in fresh complete medium − mmol/L of ingredient in cultured medium, release = mmol/L of ingredient in cultured medium − mmol/L of ingredient in fresh complete medium. The rates were reported as mmol/L per 10$^5$cells.

**Statistics.** In all cases, results are expressed as mean ± standard deviation (SD), except for Figs. 3b, 5c, d, 6a, c, 7a, d, f, 9c, 10b, d, and for Supplementary Figs. 3e, 6d, 6e and 7b (bottom) which are ± standard error (SEM). Significance was analyzed using 2-tailed Student's *t*-test, with some exceptions: Fig. 1a, b, Pearson Chi-Squared test; Figs. 1c, f, 2e, Supplementary Figs. 2e, 5d and 6c, Fisher's test; Figs. 1e, 2a, 9b, Supplementary Fig. 2a, Long-Rank test; Fig. 8b, d, Supplementary Figs. 4c–e, 5a, Wald Test; Fig. 8c, Supplementary Fig. 5d, Likelihood ratio test. When performing multiple comparison, *P*-value were adjusted for False Discovery Rate (FDR) with the Benjamini-Hochberg method.

Equal variance was assumed and the assumption was not contradicted by the data. A *P*-value of less than 0.05 was considered statistically significant. No samples or animals were excluded from analysis and sample size estimates were only used in xenograft experiments (see section Xenograft studies). Histological evaluation was performed in a blind fashion. Tumor size measurements were not blinded.

Statistical analysis of metabolomic experiments was performed using MetaboAnalyst 3.0 tool. The normalization procedure consisted of mean-centering and division by the standard deviation of each variable. The lines in the heatmap represent the relative abundance of metabolites across the samples of the two compared groups; each metabolite is indicated at the bottom of the heatmaps. The rows corresponding to the different conditions are indicated on the left. Each row corresponds to one biological. Individual samples (horizontal axis) and compounds (vertical axis) are separated using hierarchical clustering (Ward's algorithm), with the dendrogram being scaled to represent the distance between each branch (distance measure: Pearson's correlation). Statistical analyses provided were performed on Excel Office 2019 software (v17.0, Microsoft), on JMP software (v14, SAS Institute), or on Graph Pad Prism 8.

**Reporting summary.** Further information on research design is available in the Nature Research Reporting Summary linked to this article.

## Data availability

The raw and processed RNA sequencing data generated in this study have been deposited in the GEO database under accession code GSE189274. The RNA expression and associated clinicopathological data from the METABRIC cohort used in this study are available in cBioPortal under accession id brca_metabric. Human and mouse gene set annotations for pathway analysis and SGOC$_{CDK12}$ signature definition were retrieved from MSigDB [https://www.gsea-msigdb.org/gsea/msigdb/] with the msigdbr package (version 7.4.1) in R. Human GRCh38 and mouse GRCm39 reference genomes (primary genome assembly) used for the alignment of RNA sequencing data are available on GENCODE [https://www.gencodegenes.org/]. All data are available in the main text or the supplementary materials. Source data are provided with this paper.

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

## Acknowledgements

We thank the anonymous patients who donated their samples for research and the IEO Pharmacy for chemotherapy drugs. We thank R. Gunby for critically reading the manuscript; S. Pambianco, B.E. Soppo, C. Luise, D. Ricca and B. Giulini for technical assistance; A. Gobbi, M. Capillo, and the Mouse facility, and the Real Time PCR and DNA Sequencing Service of Cogentech (Cogentech Srl, Milan) for mice handling and technical support; L. Rotta and the IEO Genomic Unit for RNA sequencing experiments, the IEO Molecular Pathology Unit for tissue processing and staining; D. Disalvatore, S. Confalonieri and P. Maisonneuve for help with statistical analyses; R. Bharat (ISBE.IT) for help with metabolomics analyses; M. Monturano, G. Peruzzotti and the IEO Clinical Trial Office for assistance with clinical databases. K5-Cre mice were a gift from Dr. S. Klingler, University of Salzburg. M.G.F. was supported by a fellowship from the Fondazione Umberto Veronesi (FUV), R.P. was supported by an AIRC fellowship. This work was supported by: Associazione Italiana per la Ricerca sul Cancro (AIRC - IG 23060 and MultiUnit −5 × 1000 MCO 10.000 to P.P.D.F.; IG 11904, IG 15538, and MultiUnit -5 × 1000 MCO 10.000 to S.P.; IG 2016 to S.M.); The Italian Ministry of University and Scientific Research (MIUR) to L.A. (Italian Roadmap of European Strategy Forum on Research Infrastructures, ESFRI, to the ISBE infrastructure); The Italian Ministry of Health to D.T. (RF-2013-02358446); The Italian Ministry of University and Scientific Research (MIUR) PRIN 20177E9EPY_002 and MIUR-PRIN 202032AZT3_004 to S.P; University of Milan - PSR 2020; The Italian Ministry of Health with Ricerca Corrente and 5 × 1000 funds.

## Author contributions

Conceptualization: D.T. and S.P. Methodology: M.G.F., R.B., R.P. performed the in vitro and in vivo experiments; D.G. and M.B. did metabolomic experiments; G.B. and F.S. performed pathological analyses; G.J., M.T., F.M. did IHC stainings; M.V. contributed to the generation of the CDK12-KI mouse; A.T. performed initial in vitro and in vivo experiments; M.R. contributed with technical assistance. Formal analysis: D.G., metabolomic data analysis; E.C., S.F. and F.A.T., bioinformatic and statistical analysis on gene expression and patient data; Resources: N.B., G.Vi., G.Va., P.P.D.F., S.M., L.A., E.M., M.C., P.V. and S.P. Data curation: F.R., G.B., D.G., S.F. and F.A.T. Investigation: D.T.,

M.G.F., D.G., F.A.T., E.C., G.B., F.S. Writing and editing: D.T., and S.P. Supervision, S.P. Funding acquisition, D.T., P.P.D.F., S.M., L.A., and S.P.

## Competing interests

The authors declare no competing interests.
