## [Peer Review File · Nature Communications]

CDK12 promotes tumorigenesis but induces vulnerability to therapies inhibiting folate one-carbon metabolism in breast cancerREVIEWER COMMENTS

Reviewer #1 (Remarks to the Author):

In this interesting and timely manuscript, Tosoni and colleagues investigated a novel role of CDK12 in metabolic reprogramming contributing to CDK12-induced tumorigenesis. With a vast repertoire of molecular, mechanistic and pharmacological approaches based on in vitro, transgenic and patient derived xenograft mouse models, retrospective as well as clinical trial in vivo data, the authors demonstrate high tumor CDK12 expression creates an actionable vulnerability for breast cancer therapy.

By first establishing the oncogenic role of CDK12 on the mammary gland using epithelial-specific knock-in (CDK12-KI) mouse models, they then modelled CDK12 overexpression in non-neoplastic mammary epithelium MCF-10A. They showed that the cell phenotypes of CDK12-OE MCF10A cell fully mimicked those of CDK12-KI mammary epithelial cells (MECs), and made use of this CDK12-OE model for transcriptomic and metabolomic analysis. Noteworthy is that this uncovered interdependency between CDK12 activity as an oncogenic driver with pathological regulation of cellular metabolism, leading to identification of enhanced glycolysis and hyperactivation of the serine-glycine-one-carbon (SGOC) network as metabolic hallmarks of CDK12-induced tumorigenesis. Based on retrospective analysis of a large breast cancer patient cohort, of both adjuvant and neoadjuvant therapy data, they showed that while CDK12 high expression was a predictor of poor prognosis, more importantly it predicted more favourable response to methotrexate-based chemotherapy targeting CDK12-induced metabolic alterations. This response was also demonstrated in prospective preclinical studies using patient derived xenografts.

The study has been carefully planned, methodology sound and experiments well executed. Abundant data is presented, supportive of the findings. Of conceptual significance is the association between CDK12 oncogenic function with pathological metabolic regulation, highlighting the importance of deeper understanding of metabolic vulnerabilities to aid identification of targeted anti-cancer therapies. The findings are novel and highly relevant for translational consideration.

There are a few minor concerns that should be addressed:

Page 9. Line 166-167 "...while the opposite effects were observed in control MCF10A cells (Fig 4A). This statement could be misleading. Rather than the opposite effect, the EV cells showed no response to decreasing concentrations of glucose, nor to addition of glucose inhibitor 2DG.

Page 13. Line 260-261 "These results indicate that CDK12 HIGH patients show selective sensitivity to CMF therapy compared with CDK12 LOW and CDK12 NULL patients,....." Indeed Figure 6 A which shows CDK12 HIGH and CDK12 LOW, and supplementary Figure 5B, which shows CDK12 NULL, do show significant difference of response between CMF vs Tax/AC for CDK12 HIGH and not for the other two. However, it is supplementary Figure 5A, right hand panel which introduces some confusion. In this panel which shows HER2-negative patients treated with CMF, HR HIGH vs LOW shows no significant difference, $p = 0.44$. This means there is no significant difference in response to CMF between CDK12 HIGH and CDK12 LOW patients.

Supplementary methods, page 2. Line 29—30. "...as the CDK12 NULL group invariably showed poor prognosis and therapy resistance independently of the type of chemotherapy

regimen.....” is not quite accurate a statement, nor sufficient reason why the CDK12 NULL group was excluded from further study. Supplementary Figure 4 C and D both show the blue and red lines close to each other, ie. CDK12 HIGH and CDK12NULL both showing poor prognosis. Likewise for Supplementary Fig 5A, left panel HER2-neg patients treated with Tax/AC treatment, the red and blue lines also overlap.

It would be most helpful if the authors could include a flow chart or diagram that summarizes the concepts and relevance.

Reviewer #2 (Remarks to the Author):

This manuscript provides evidence that CDK12 over-expression is oncogenic through metabolic rewiring. It is suggested that this provides a therapeutic vulnerability through sensitization to methotrexate. This an excellent well performed study. I had just a few suggestions that may improve the manuscript.

1. Evidence that human CDK12 is actually (over)-expressed in the knock-in mouse mammary gland and in the tumors that develop needs to be provided (IHC or western or RNA). Preferably the relative expression should be compared to endogenous mouse CDK12.
2. The difference in fig 2A appears marginal whereas in B and C numbers of tumors and size are clearly different. What is the explanation for this? It is certainly worth a comment in the text.
3. Mechanism: Given the changes in mRNA levels it appears the effect is transcriptional. How do the authors imagine these particular genes are dysregulated? Are there specific properties of the promoters of these genes? eg Is intronic polyadenylation differentially affected as is seen in CDK12 KO (Dubbury et al (2018) Nature)?

Reviewer #3 (Remarks to the Author):

NCOMMS-21-35816

In this study, Tosoni et al. study the relevance of CDK12 for breast tumor promotion and therapy-response to family of antimetabolic drugs. The study covers from the causal contribution of CDK12 to disease pathogenesis using primary cultures, genetic mouse models, cellular systems and PDX, to clinical data supporting the prognostic and predictive relevance. They accompany the results with molecular studies that provide a feasible explanation and reveal therapeutic opportunities for CDK12-based patient stratification. This represents an excellent study, with molecular, preclinical and clinical data that are solid and conclusive. I consider the study highly relevant and appropriate for publication in this journal. There are only a few minor comments or suggestions for the authors to consider.

1. My main aspect to point out is that I do not understand the association of both CDK12 High AND Low to BCa prognosis, while the whole manuscript is suggesting a pro-tumorigenic role. This would suggest that the CDK12-KD cells could have some aggressive phenotype as well.
2. Also important, I do not understand the clinical association of CDK12 High to TaxAC refractoriness (S5D), as the authors make specific emphasis on the relevance for MTX-sensitivity, but do not show TaxAC association in prior clinical studies in the manuscript. This is also suggested in Fig. 8A with PTX.

3. I would suggest including information relative to breast cancer pathological, epidemiological and molecular characteristics in the intro.
4. The rationale for using a CK5 promoter (basal) should be better explained.
5. The authors could comment on the pathological abnormalities of the WT mice in the model presented in figure 1.
6. The mechanism of regulation of the SGOC genes by CDK12 remains elusive, as the authors mention in the discussion. It would be interesting to explore it, but I consider that it is out of the scope of this study in the current form and content.

Stats:

7. There are stats missing in some panels (S3E, 7C).
8. Some of the survival curves in Fig. 7B appear too close to present statistically significant differences.
9. I would consider the correlation of the SGOC signature in TNBC (Fig 6A) modest, despite significant (the p-value in correlations is strongly influenced by the number of cases)

POINT BY POINT REPLY TO THE REVIEWERS' COMMENTS (reproduced verbatim)

Reviewer #1

In this interesting and timely manuscript, Tosoni and colleagues investigated a novel role of CDK12 in metabolic reprogramming contributing to CDK12-induced tumorigenesis. With a vast repertoire of molecular, mechanistic and pharmacological approaches based on in vitro, transgenic and patient derived xenograft mouse models, retrospective as well as clinical trial in vivo data, the authors demonstrate high tumor CDK12 expression creates an actionable vulnerability for breast cancer therapy.

By first establishing the oncogenic role of CDK12 on the mammary gland using epithelial-specific knock-in (CDK12-KI) mouse models, they then modelled CDK12 overexpression in non-neoplastic mammary epithelium MCF-10A. They showed that the cell phenotypes of CDK12-OE MCF10A cell fully mimicked those of CDK12-KI mammary epithelial cells (MECs), and made use of this CDK12-OE model for transcriptomic and metabolomic analysis. Noteworthy is that this uncovered interdependency between CDK12 activity as an oncogenic driver with pathological regulation of cellular metabolism, leading to identification of enhanced glycolysis and hyperactivation of the serine-glycine-one-carbon (SGOC) network as metabolic hallmarks of CDK12-induced tumorigenesis. Based on retrospective analysis of a large breast cancer patient cohort, of both adjuvant and neoadjuvant therapy data, they showed that while CDK12 high expression was a predictor of poor prognosis, more importantly it predicted more favourable response to methotrexate-based chemotherapy targeting CDK12-induced metabolic alterations. This response was also demonstrated in prospective preclinical studies using patient derived xenografts.

The study has been carefully planned, methodology sound and experiments well executed. Abundant data is presented, supportive of the findings. Of conceptual significance is the association between CDK12 oncogenic function with pathological metabolic regulation, highlighting the importance of deeper understanding of metabolic vulnerabilities to aid identification of targeted anti-cancer therapies. The findings are novel and highly relevant for translational consideration.

R. We thank Reviewer #1 for the appreciative words and for highlighting the novelty of our findings linking CDK12 oncogenic function to pathological regulation of the cellular metabolism, and their intrinsic translational relevance.

This Reviewer raised a few minor concerns that we address below (*the different points are reproduced verbatim and numbered according to the original reviewer's points for convenience*):

1) Page 9. Line 166-167 "...while the opposite effects were observed in control MCF10A cells (Fig 4A). This statement could be misleading. Rather than the opposite effect, the EV cells showed no response to decreasing concentrations of glucose, nor to addition of glucose inhibitor 2DG.

R. Agree. We have modified the text to make it clear that, in contrast with CDK12-OE MCF10A cells, control MCF10A cells are insensitive to glucose deprivation or 2-DG treatment, while being sensitive to serine deprivation (page 10, lines 194-196).

2) Page 13. Line 260-261 "These results indicate that CDK12 HIGH patients show selective sensitivity to CMF therapy compared with CDK12 LOW and CDK12 NULL patients,....."

Indeed Figure 6 A which shows CDK12 HIGH and CDK12 LOW, and supplementary Figure 5B, which shows CDK12 NULL, do show significant difference of response between CMF vs Tax/AC for CDK12 HIGH and not for the other two. However, it is supplementary Figure 5A, right hand panel which introduces some confusion. In this panel which shows HER2-negative patients treated with CMF, HR HIGH vs LOW shows no significant difference, $p = 0.44$. This means there is no significant difference in response to CMF between CDK12 HIGH and CDK12 LOW patients.

R. Agree. We acknowledge the Reviewer's point that it might generate some confusion to compare the clinical data in the original Fig. 6B with those shown in the original Suppl. Fig. 5A. In response to this point, we would like to clarify that these two figures, although based on the same set of results, convey complementary information:

2a) In the original Fig. 6B, CDK12-HIGH and CDK12-LOW patients were independently analyzed for their response to Tax/AC vs. CMF. In the revised version of Fig. 6b, we now include the CDK12-NULL group of patients (originally shown in Suppl. Fig. 5B) to provide a comprehensive view of how the three CDK12 categories behave in their response to Tax/AC vs. CMF:

- CDK12-HIGH patients, who have an overall adverse prognosis (shown in the old Suppl. Fig. 4C, now the revised Suppl. Fig. 4c), are the only patients that respond with a significant reduction in the rate of distant metastasis when treated with CMF, compared to Tax/AC (new Fig. 6b, left panel). This finding implies that while CDK12 overexpression is an overall marker of adverse prognosis, it also represents a predictive biomarker of selective response to CMF in CDK12-HIGH patients.

- CDK12-LOW patients, who have an overall good prognosis (shown in the old Suppl. Fig. 4C, now the revised Suppl. Fig. 4c), do not show any further prognostic improvement, irrespective of Tax/AC or CMF treatment (new Fig. 6b, middle panel); this finding implies that CDK12 cannot serve as a marker predictive of chemotherapy response in CDK12-LOW patients.

- CDK12-NULL patients, who have an overall adverse prognosis (shown in the old Suppl. Fig. 4C, now the revised Suppl. Fig. 4c) maintain an adverse disease course in response to CMF as well as Tax/AC (new Fig. 6b, right panel). The aggressive disease course of CDK12-NULL patients is in keeping with previous reports describing aggressive disease in loss-of-function CDK12 patients (see also response to point 3 below). Thus, the biological aggressiveness of CDK12-NULL tumors is likely driven by mechanisms linked to loss-of-CDK12 function, which are however different from those underlying CDK12 overexpression-driven tumorigenesis that are responsible for the selective vulnerability of CDK12-HIGH patients to CMF.

The Reviewer is therefore correct noting that:

- only the aggressive CDK12-HIGH patients show prognostic improvement when treated with CMF, while CDK12-LOW and CDK12-NULL patients maintain a good or poor prognosis, respectively, independently of chemotherapy treatment.

- there is no significant difference between CDK12-HIGH and CDK12-LOW patients in response to CMF ($p=0.44$), as the prognostic behavior of CDK12-HIGH patients upon CMF treatment improves dramatically (distant recurrence rate goes from 30.4% to 12.2%), thus reaching that observed in CDK12-LOW patients (11%).

Considering the above, and also this Reviewer's suggestion in point 5 below, we have included a box diagram summarizing all these findings in the new Suppl. Fig. 5b, to provide a more immediate visualization of the prognostic outcome of the three CDK12 groups in response to the different therapies (Tax/AC vs. CMF).

2b) In the old Suppl. Fig. 5A, we provided a simultaneous view of the response of the three CDK12 categories to Tax/AC and to CMF, showing that:

- In response to Tax/AC (left panel of the old Suppl. Fig. 5A, now top panel of the new Suppl. Fig. 5a), CDK12-HIGH patients show the most adverse disease outcome, similar to CDK12-NULL patients, while CDK12-LOW patients show the most favorable disease course.

- In response to CMF (right panel of the old Suppl. Fig. 5A, now bottom panel of the new Suppl. Fig. 5a), CDK12-HIGH patients show a dramatically improved disease course that becomes as favorable as that observed in CDK12-LOW patients (as correctly pointed out by the Reviewer). In contrast, CDK12-NULL patients still have a poor disease outcome.

We have modified the text in the revised manuscript to better explain the above set of results (see highlighted changes in lines 284-299, pages 13-14).

3) Supplementary methods, page 2. Line 29—30. “...as the CDK12 NULL group invariably showed poor prognosis and therapy resistance independently of the type of chemotherapy regimen.....” is not quite accurate a statement, nor sufficient reason why the CDK12 NULL group was excluded from further study.

R. We acknowledge the Reviewer’s point. We would like to better explain why we have not performed a thorough characterization of CDK12 NULL tumors through functional studies, nor included them as controls in studies assessing the specificity of the response of CDK12-HIGH patients to methotrexate-based therapy:

a) CDK12-NULL patients have an overall adverse aggressive prognosis (shown in the old Suppl. Fig. 4C, now Suppl. Fig. 4c).

b) The aggressive disease course of CDK12-NULL patients showed no variation upon stratification by chemotherapy type. This was shown in the old Suppl. Fig. 5B (now Fig. 6b in the main text), where the Kaplan-Meier curves representing Tax/AC-treated vs. CMF-treated CDK12-NULL patients completely overlapped, indicating that the disease course of CDK12-NULL patients is not influenced by chemotherapy type.

c) Based on integrative IHC and QPCR analysis of CDK12 protein and mRNA expression (see Suppl. Methods, “Patient cohort and stratification”), we submit that CDK12-NULL tumors, with not-overexpressed CDK12 protein by IHC and undetectable CDK12 mRNA expression by QPCR, correspond to breast cancers previously described to bear CDK12 loss-of-function which, likewise ovarian and prostate cancers with dysfunctional CDK12 status, typically display an aggressive disease course (Naidoo, K. et al., Mol Cancer Ther, 2018; Quereda, V. et al., Cancer Cell, 2019; Wu, Y. M. et al., Cell, 2018; Popova, T. et al., Cancer Res, 2016).

Therefore, based on the intrinsic poor prognosis of CDK12-NULL patients, which is not influenced by chemotherapy type, we excluded these patients from further functional studies. As a control for the selective vulnerability of CDK12-HIGH tumors to CMF, we decided to use the CDK12-LOW group since they express similar CDK12 mRNA levels to CDK12-HIGH tumors but not overexpression of the CDK12 protein.

4) Supplementary Figure 4 C and D both show the blue and red lines close to each other, ie. CDK12 HIGH and CDK12NULL both showing poor prognosis. Likewise for Supplementary Fig 5A, left panel HER2-neg patients treated with Tax/AC treatment, the red and blue lines also overlap.

R. Agree. This point is related to the point 2 and we refer the Reviewer to our detailed reply above, where we describe the prognosis and therapy response of the different CDK12 groups and clarify that:

- In the old Suppl. Fig. 4 C and D (now Suppl. Fig. 4c,d), we describe the overall prognostic behavior of the different CDK12 groups (regardless of chemotherapy type), showing that both CDK12-NULL and CDK12-HIGH patients have unfavorable prognosis, as opposed to the favorable prognosis of CDK12-LOW patients.
- In the old Suppl. Fig. 5A (now Suppl. Fig. 5a), we describe the response of CDK12-HIGH and CDK12-NULL to Tax/AC (left hand panel of the old figure) or to CMF (right hand panel of the old figure). As correctly observed by this Reviewer, this figure shows that CDK12-HIGH and CDK12-NULL patients are equally resistant to Tax/AC treatment (witnessed by the intercrossing of the blue and red lines that represent the recurrence rates of these two groups). In contrast, in response to CMF (right hand panel of the old figure), CDK12-NULL patients retain an adverse prognosis due to refractoriness to treatment, while CDK12-HIGH patients show a dramatically improved disease course that overlaps the favorable disease course observed in CDK12-LOW patients.

5) It would be most helpful if the authors could include a flow chart or diagram that summarizes the concepts and relevance.

R. Agree. Following the Reviewer's suggestion, we have included a new Suppl. Fig. 5b showing a schematic box diagram summarizing the prognostic outcome of patients, stratified by CDK12 status, in response to the different types of chemotherapies.

Reviewer #2

This manuscript provides evidence that CDK12 over-expression is oncogenic through metabolic rewiring. It is suggested that this provides a therapeutic vulnerability through sensitization to methotrexate. This an excellent well performed study. I had just a few suggestions that may improve the manuscript.

R. We thank Reviewer #2 for the appreciative words and for the suggestions provided, which helped us to craft an improved version of our manuscript.

1) Evidence that human CDK12 is actually (over)-expressed in the knock-in mouse mammary gland and in the tumors that develop needs to be provided (IHC or western or RNA).

Preferably the relative expression should be compared to endogenous mouse CDK12.

R. We acknowledge the relevance of the Reviewer's point. To address this point, we have used an anti-CDK12 monoclonal antibody (included also in Methods; see line 563 of the revised version) with a dual mouse and human cross-reactivity to detect: *i*) endogenous CDK12 expression in the mammary gland of WT mice; *ii*) transgenic human CDK12 expression in both preneoplastic and overtly neoplastic lesions in the mammary epithelium of CDK12 knock-in mice. We have also included in this analysis a xenograft of the CDK12-overexpressing mammary human epithelial cell line, BT474, to show that transgenic CDK12 expression in the knock-in mouse mammary gland is comparable to that observed in CDK12-overexpressing human breast cancer cells. These results are shown in the new Suppl. Fig. 1 and described in the revised version of the manuscript (see Results, lines 73-78).

2) The difference in fig 2A appears marginal whereas in B and C numbers of tumors and size are clearly different. What is the explanation for this? It is certainly worth a comment in the text.

R. Agree. The apparently marginal, albeit statistically significant difference in tumor-free survival between PyMT-WT and PyMT-CDK12 mice can be explained by the accelerated tumor progression kinetics of WT female mice in the PyMT breast cancer model (see for instance, Williams T.M. et al. JBC, 2004). We became aware of this problem when we performed these experiments, observing also that the difference in tumor-free survival between PyMT-WT vs. PyMT-CDK12 male mice was far greater than that observed in female mice, likely because of the slower progression kinetics. We have now included these results with male mice in the new Fig. 2 and added an explanation in the text relative to this experiment (see Results, page 6, lines 111-114).

Concerning the apparent discrepancy between the survival curves and the number/size of tumors between PyMT-WT vs. PyMT-CDK12 mice, we would like to thank the Reviewer for bringing to our attention that we had omitted to clearly describe how we scored the differences in tumor burden reported in the old Fig. 2A. In preliminary experiments, because of the accelerated tumor progression kinetics of the PyMT model, we noted that the differences in the tumor burden between PyMT-WT vs. PyMT-CDK12 female mice were attenuated when animals were analyzed at >7-8 weeks of age. Therefore, in addition to experiments performed to evaluate tumor-free survival, we used an independent set of female PyMT-WT vs. PyMT-CDK12 mice (described in the old Fig. 2A, now the new Fig. 2a) to analyze tumor size and number at a fixed time-point of approximately 5 weeks of age, corresponding to the time when the tumors in the control PyMT-WT mice had reached a palpable size. We have now included this detail in the legend to the new Fig. 2.

3) Mechanism: Given the changes in mRNA levels it appears the effect is transcriptional. How do the authors imagine these particular genes are dysregulated? Are there specific properties of the promoters of these genes? eg Is intronic polyadenylation differentially affected as is seen in CDK12 KO (Dubbury et al (2018) Nature)?

R. Agree. In the original version of the manuscript (see page 19, lines 408-413), we state that the mechanisms behind the transcriptional downregulatory control exerted by CDK12 over several metabolic enzymes of the serine-one-carbon metabolism remains an open question. At present, we can only speculate about the existence of a mechanism that allows CDK12 to regulate *en bloc* a set of metabolic genes likely by interfering with an upstream signaling circuitry involved in their regulation or through CDK12-dependent epigenetic modifications. While we are actively pursuing these two lines of investigations, this will take considerable time and will hopefully represent the object of a future publication.

Reviewer #3

In this study, Tosoni et al. study the relevance of CDK12 for breast tumor promotion and therapy-response to family of antimetabolic drugs. The study covers from the causal contribution of CDK12 to disease pathogenesis using primary cultures, genetic mouse models, cellular systems and PDX, to clinical data supporting the prognostic and predictive relevance. They accompany the results with molecular studies that provide a feasible explanation and reveal therapeutic opportunities for CDK12-based patient stratification. This represents an excellent study, with molecular, preclinical and clinical data that are solid and conclusive. I consider the study highly relevant and appropriate for publication in this journal. There are only a few minor comments or suggestions for the authors to consider.

R. We are most thankful to Reviewer #3 for the appreciative words on our study.

This Reviewer had a few minor comments and suggestions that we have systematically addressed (*the different points are reproduced verbatim and numbered according to the original reviewer's points for convenience*):

1) My main aspect to point out is that I do not understand the association of both CDK12 High AND Low to BCa prognosis, while the whole manuscript is suggesting a pro-tumorigenic role. This would suggest that the CDK12-KD cells could have some aggressive phenotype as well.

R. We acknowledge this Reviewer's point. This point concerns *i)* the differences in the default overall prognostic behavior of the different CDK12 groups and *ii)* the changes in their prognostic outcome when stratified by response to chemotherapy, namely Tax/AC vs. CMF. We would like to refer this Reviewer to our reply to a similar point raised by Reviewer 1 (see point #2 and #4 of Reviewer 1 above), where we describe the prognostic behavior of the three CDK12 categories (i.e., good prognosis for CDK12-LOW and adverse prognosis for both CDK12-HIGH and CDK12-NUL patients; see also the old Suppl. Fig. 4C, now the new Fig. 4c, and how these three categories respond to Tax/AC or CMF treatment (see also the box diagram included in the new Suppl. Fig. 5b of the revised version).

2) Also important, I do not understand the clinical association of CDK12 High to TaxAC refractoriness (S5D), as the authors make specific emphasis on the relevance for MTX-sensitivity, but do not show TaxAC association in prior clinical studies in the manuscript. This is also suggested in Fig. 8A with PTX.

R. Acknowledged. In response to this point, we would like to note that the association between CDK12-HIGH status and resistance to Tax/AC chemotherapy is based on a dual observation:

a) The analysis of the adjuvant cohort (see old Suppl. Fig. 5A, left hand panel, now top panel in the new Suppl. Fig. 5a) showed that CDK12-HIGH patients treated with Tax/AC have a poor prognosis in contrast to CDK12-LOW patients who show a more favorable prognosis.

b) The analysis of the neoadjuvant cohort (see old Suppl. Fig. 5D, now included as new Suppl. Fig. 5d) in which CDK12-HIGH patients resisted Tax/AC treatment, with reduced probability of undergoing complete tumor remission (defined as "pathological complete response", PCR) compared to CDK12-LOW patients.

Of note, the association between CDK12-HIGH status and resistance to Tax/AC treatment, as correctly observed by this Reviewer, was also confirmed in the *in vivo* analysis of tumor growth reported in the old Fig. 8A (now the revised Suppl. Fig. 8a).

For a schematic representation of the differences in the prognostic outcome of the different CDK12 patient groups in response to the different types of chemotherapy (i.e., Tax/AC and CMF), we would like to refer this Reviewer to the schematic in the new Suppl. Fig. 5b.

3) I would suggest including information relative to breast cancer pathological, epidemiological and molecular characteristics in the intro.

R. Acknowledged. We have included in the revised Introduction a description of the current taxonomy for breast cancer, including the differences in terms of epidemiology and therapeutic options for the different molecular subtypes of breast cancer (see page 3, lines 17-31).

4) The rationale for using a CK5 promoter (basal) should be better explained.

R. Agree. We have included at the beginning of the revised Results a better explanation of the rationale for using the basal CK5 promoter. We now better explain that, based on our evidence that endogenous CDK12 is expressed both in the basal and luminal mammary epithelial layers in homeostatic conditions (as shown in the old Suppl. Fig. 1A, now Suppl. Fig. 1b), the use of the CK5 promoter represented a suitable strategy to achieve early transgenic CDK12 overexpression in the entire mammary epithelium, starting from the basal layer that generates the suprabasal layer. Instead, the use of the MMTV promoter, which is more commonly used for the targeted overexpression of transgenes in the mouse mammary gland, would have resulted in a delayed CDK12 overexpression, since the MMTV promoter is influenced by the presence of estrogens that appear after three weeks of age, and would have also been confined to the luminal, estrogen-sensitive layer of the mammary epithelium.

5) The authors could comment on the pathological abnormalities of the WT mice in the model presented in figure 1.

R. We acknowledge this Reviewer's point. We note that the appearance of pathological abnormalities in the mammary epithelium is a well-established occurrence in WT FVB mice. In particular, while the appearance of benign hyperplastic lesions can be observed in young adult mice, the spontaneous acquisition of preneoplastic and overtly neoplastic lesions is a well-known age-related occurrence, which becomes increasingly prevalent starting at 12-13 months of age and is often associated with other age-related pathological manifestations (see, for instance, Mahler JF, *et al.*, Spontaneous lesions in aging FVB/N mice, *Toxicol Pathol.* 24(6):710-6, 1996; Radaelli E., *et al.*, Mammary tumor phenotypes in wild-type aging female FVB/N mice with pituitary prolactinomas. *Vet Pathol.*, 46(4):736-45, 2009; Raafat A., *et al.* Effects of age and parity on mammary gland lesions and progenitor cells in the FVB/N-RC mice. *PLoS One*, 7(8): e43624, 2012).

We would also like to note that the type and frequency of age-related abnormalities in our 12-24-month old WT mice is in keeping with previous reports in the literature, in particular as regards the lobuloalveolar hyperplastic/preneoplastic aberrations and the peculiar squamous metaplasia associated with infiltrating neoplastic lesions (Lalage M Wakefield *et al.*, Spontaneous pituitary abnormalities and mammary hyperplasia in FVB/NCr mice: implications for mouse modelling. *Comp Med*, 53(4):424-32, 2003). Furthermore, we refer this Reviewer to the detailed description of the frequency and thorough histological characterization of the lesions that we observed in WT as well as in CDK12-knock-in mice reported in Source data.

6) The mechanism of regulation of the SGOC genes by CDK12 remains elusive, as the authors mention in the discussion. It would be interesting to explore it, but I consider that it is out of the scope of this study in the current form and content.

R. We thank the Reviewer for this insightful thought. We refer this reviewer to our response to a similar point raised by Reviewer #2 (point 3), where we described the lines of investigation that we are actively pursuing with the aim to shed light on the CDK12-induced regulation of SGOC genes.

7) There are stats missing in some panels (S3E, 7C).

R. Acknowledged. We have included the statistics in the two figures indicated by the Reviewer in the revised version. Raw data for all the independent experiments are reported in the source data files submitted with the revised version of the manuscript.

8) Some of the survival curves in Fig. 7B appear too close to present statistically significant differences.

R. Acknowledged. We have double checked the statistics, confirming the data included in the original version. We provide all the raw data in the source data file.

9) I would consider the correlation of the SGOC signature in TNBC (Fig 6A) modest, despite significant (the p-value in correlations is strongly influenced by the number of cases)

R. Acknowledged. We are aware that the p-value in TNBC is much lower than those obtained with the other molecular subtypes. We note, however, that, according to extant literature, TNBCs are typically metabolically very active and represent a paradigm for the deregulation of the serine-one-carbon metabolism, arguing that several mechanisms, other than CDK12 overexpression, may directly or indirectly converge on the deregulation of this metabolic pathway (see for instance, Possemato R. *et al.*, Functional genomics reveal that the serine synthesis pathway is essential in breast cancer. Nature, 2011).

Supporting this notion, we also found that TNBCs display a higher basal level of the SGOC signature score, compared to the other molecular subtypes (shown in the old Fig. 6A, new Fig. 6a). Therefore, despite the higher activation status of the SGOC metabolism in TNBCs, our observation that a correlation between high CDK12 mRNA expression and the SGOC gene signature still holds in these tumors, makes this correlation even more significant.

REVIEWER COMMENTS

Reviewer #1 (Remarks to the Author):

The authors have adequately addressed the previous comments from this reviewer and I have no further points to add.

Reviewer #2 (Remarks to the Author):

I am content that my questions/concerns have been addressed appropriately.

Reviewer #3 (Remarks to the Author):

The authors have addressed my remaining comments. Due to the complexity of the prognostic vs. predictive potential of CDK12, I strongly suggest that the authors make an additional effort to convey clearly the results and the conclusions of the clinical analyses, integrated with the molecular studies. The concept of reverting the aggressive behavior in CDK12-High (bringing it closer to CDK12-LOW) is interesting, as it reveals an oncogene addiction-like therapeutic vulnerability.

POINT BY POINT REPLY TO THE REVIEWERS' COMMENTS (reproduced verbatim)

Reviewer #1

The authors have adequately addressed the previous comments from this reviewer and I have no further points to add.

Reviewer #2

I am content that my questions/concerns have been addressed appropriately.

R. We are glad to hear that Reviewer #1 and #2 were satisfied with our efforts to address all their questions and concerns and would like to take this occasion to thank them once again for all the insightful suggestions that helped us to craft an improved version of our manuscript.

Reviewer #3

The authors have addressed my remaining comments. Due to the complexity of the prognostic vs. predictive potential of CDK12, I strongly suggest that the authors make an additional effort to convey clearly the results and the conclusions of the clinical analyses, integrated with the molecular studies. The concept of reverting the aggressive behavior in CDK12-High (bringing it closer to CDK12-LOW) is interesting, as it reveals an oncogene addiction-like therapeutic vulnerability.

R. The request from this Reviewer to make an additional effort to better convey results from patient cohort studies on the dual clinical value of CDK12 as a clinical biomarker of poor prognosis while behaving as a predictor of positive response to targeted antimetabolite therapy against the pathological metabolisms linked to CDK12-induced tumorigenesis was very well taken.

We also greatly appreciated the suggestion of the Reviewer to better point out the concept of “*oncogene addiction-like therapeutic vulnerability*”, which we have adopted in several parts of the manuscript. Towards the line of thoughts of this Reviewer, we also accepted the new title proposed by the Editorial Committee “*CDK12 promotes tumorigenesis but induces vulnerability to therapies inhibiting folate one-carbon metabolism in breast cancer*”, which we believe better conveys the message that CDK12, while being a strong driver of tumorigenesis, introduces a metabolic Achille’s heel in CDK12-overexpressing breast tumors, thereby providing a unique therapeutic option for otherwise poor prognosis and therapy refractory cancers.

Listed below are the changes that we have introduced in the new revised version of our manuscript to address the Reviewer’s request:

- change of the subheading of the paragraph reporting the results from clinical cohort studies, along the line of the title proposed by the Editorial Committee for the entire manuscript (please, see page 12, line 235 of the revised version);
- a few changes (all highlighted in yellow) in the text of the paragraph reporting the clinical results that we believe aid a better understanding of the experimental design (please, see in the revised version page 12, lines 252-253; page 13, line 265, lines 272-274, 279, 280; page 14, lines 283-286 and 295-298);
- a new table, included as Supplementary Fig. 5e, reporting a schematic overview of the clinical value of CDK12 as a molecular biomarker suitable for patient stratification for prognosis and

for chemotherapy response prediction (described at page 15-16, lines 330-332 of the main text of the revised version), based on the bulk of clinical evidence from patient cohort studies described in the new Fig. 8 and in the new Supplementary Fig. 5;

- a sentence in the Discussion (please, see page 20, lines 429-433) which highlights the effect of methotrexate-based therapy in reverting the poor prognosis of CDK12^{HIGH} patients to a more favorable disease course vis a vis the absence of any significant effect of this treatment in CDK12^{LOW} or CDK12^{NULL} patients, to address the fundamental concept of “oncogene addiction-like therapeutic vulnerability” associated with CDK12 overexpression in human breast cancers.